# The COMPASS subunit Spp1 protects nascent DNA at the Tus/*Ter* replication fork barrier by limiting DNA availability to nucleases

Nagham Ghaddar [1], Yves Corda [1], Pierre Luciano [1], Martina Galli[2], Ylli Doksani [2] & Vincent Géli [1] ✉

Homologous recombination factors play a crucial role in protecting nascent DNA during DNA replication, but the role of chromatin in this process is largely unknown. Here, we used the bacterial Tus/*Ter* barrier known to induce a site-specific replication fork stalling in *S. cerevisiae*. We report that the Set1C subunit Spp1 is recruited behind the stalled replication fork independently of its interaction with Set1. Spp1 chromatin recruitment depends on the interaction of its PHD domain with H3K4me3 parental histones deposited behind the stalled fork. Its recruitment prevents the accumulation of ssDNA at the stalled fork by restricting the access of Exo1. We further show that deleting SPP*1* increases the mutation rate upstream of the barrier favoring the accumulation of microdeletions. Finally, we report that Spp1 protects nascent DNA at the Tus/*Ter* stalled replication fork. We propose that Spp1 limits the remodeling of the fork, which ultimately limits nascent DNA availability to nucleases.

During DNA replication, the replisome must unwind the DNA double helix, ensure faithful DNA duplication as well as deal with any impediment it may encounter[1]. Alteration of replication fork (RF) progression, defined generally as replication stress, is caused by numerous mechanisms[2]. For instance, the replisome could stall and collapse when colliding with transcriptional machinery or facing a Replication Fork Barrier (RFB) such as protein bound-DNA or repetitive sequences[3]. These challenges, if not regulated, can lead to DNA breaks driving genomic instability and cancer development[4–6]. Cells have adapted to cope with such challenges through the collaborative work between replisome components, fork repair machinery and cell cycle-dependent kinases, ensuring proper replication resumption[7]. Several fork-associated repair mechanisms promote fork recovery by bypassing obstacles such as translesion synthesis and re-priming activities, template switch, break-induced replication and homologous recombination[7–10].

The regulation of the choice of repair processes is still to be fully understood[2,8,10–12]. Stalled replication forks can be processed by resection nucleases such as Exo1[EXO1]/Dna2[DNA2]/Mre11[MRE11] and remodelers such as MRX[MRN], Mph1[FANCM]/Rad5, RAD51 while protecting the nascent DNA

strands to allow fork recovery[11,12]. A key component to nascent DNA protection is RPA-coated ssDNA that can be displaced by Rad52 and allow Rad51[RAD51] loading to the nascent DNA[13,14]. The stabilized fork can then be rescued by downstream forks or by recombination[9,11,14].

DNA replication occurs in a crowded chromatin environment where the replisome itself can disrupt chromatin organization[15–17]. Histone remodelers and chaperones aid the replisome in nucleosome disassembly ahead of the fork and reassembly behind the fork[18]. The nascent chromatin contains a mix of recycled parental histones (marked by H3K4me3) and newly synthesized histones (marked by H3K56ac)[19,20]. While chromatin remodelers such as INO80, SWI/SNF, Fun30, and RSC were shown to remodel the chromatin surrounding a double-strand break (DSB) to allow resection[21–25], how replication stress responses and repair mechanisms are shaped by the chromatin environment is still to be fully understood[26].

In budding yeast, all patterns of H3K4 methylation (mono-, di-, and tri-) are deposited by Set1 histone methyltransferase, which belongs to an evolutionarily conserved complex (called Set1C or COMPASS). The Set1 subunit of Set1C acts as a scaffold for seven additional subunits (Swd1, Swd2, Swd3, Bre2, Sdc1, Shg1, and

[1]Marseille Cancer Research Centre (CRCM), U1068 INSERM, UMR7258 CNRS, UM105 Aix-Marseille University, Institute Paoli-Calmettes, Ligue Nationale Contre le Cancer (Equipe Labellisée), Marseille, France. [2]IFOM ETS - the AIRC Institute of Molecular Oncology, Milan, Italy. ✉e-mail: vincent.geli@inserm.fr

Spp1)[27–30]. While the absence of Set1 affects all states of H3K4 methylation, inactivation of Spp1, the PHD finger domain-containing subunit, affects only H3K4me3[30,31]. Set1C was initially shown to be involved in DSB repair by NHEJ[32]; recent data indicate that Set1-dependent H3K4 methylation acts as a decelerator for replisome progression at highly transcribed genes to prevent Transcription-Replication conflicts (TRC)[33] and limits DNA damage in response to changes in S-phase dynamics[34]. Moreover, Set1C was shown to act in parallel with Gcn5 and MRX at arrested forks to increase chromatin accessibility and allow fork recovery[35].

Similarly, in the context of BRCA-deficient mammalian cells, H3K4 methylation catalyzed by MLL3/4 promotes the resection of stalled forks[36]. In contrast, the mammalian SET1DA with BOD1L protects nascent DNA degradation by promoting FANCD2 chaperone activity and inhibiting chromatin remodeler activities at the stalled forks[37]. These results underscore the need to understand how the Set1C complex is recruited to stalled forks and the role it plays in the choice of the replication stress response.

The Tus/Ter barrier system of *E. coli* consists of 21-bp DNA sequences (Ter) bound by the terminator protein Tus that can block replication forks unidirectionally[38]. This system has been used as a site-specific replication fork barrier in yeast and mammalian cells[39,40]. Here, we used the Tus/Ter system to study how Set1C contributes to the replication stress response at a unidirectional, site-specific replication barrier[41]. We show that the Spp1 subunit of Set1C is recruited via its PHD domain to the Tus/Ter stalled fork independently of Set1C. Its recruitment prevents the formation of excessive ssDNA upon replication stress and limits DNA availability to Exo1-mediated resection. Our results indicate that Spp1 protects nascent DNA when the fork is stalled at the Tus/Ter barrier. We propose a model in which Spp1 binding to methylated histones behind a stalled replication fork promotes protective nascent chromatin, thus limiting remodeling of the fork and deleterious ssDNA accumulation.

## Results

### The Set1C subunit Spp1 is recruited to Tus/Ter replication fork barrier

To assess whether Set1C has a role at stalled forks, we used the previously described galactose inducible and site-specific Tus/Ter replication fork barrier[41,42]. We introduced 21 arrays of the *TerB* sequence in the restrictive orientation downstream of ARS305, an early origin of replication, where *URA3* serves as a reporter gene upstream of the barrier (Fig. 1a). Cells were synchronized in G1 with α-factor and then released into S phase in galactose-rich media to sustain Tus gene expression. In agreement with previous findings[42,43], using two-dimensional agarose gel electrophoresis (2D-gels), we found that the 21xTus/Ter replication fork barrier (RFB) efficiently but transiently stalls the replication forks (Fig. 1b). We also detected a visible accumulation of X-shaped DNA intermediates, which could be attributed to replication fork reversal, recombination intermediates, or converging forks arriving from the ARS306 origin. To monitor the replisome progression, we measured by chromatin immunoprecipitation (ChIP) the occupancy of the Cdc45 subunit of the Cdc45-Mcm2/7-GINS (CMG) replicative helicase complex. The ChIP of Cdc45-V5 was performed at several time points in the S phase (Fig. 1c). Expression of Tus and progression through the cell cycle were monitored by WB and FACS, respectively (Supplementary Fig. 1a, b). Cdc45-V5 occupancy was quantified using qPCR with a pair of primers covering the 9th and 10th *Ter* repeats (Fig. 1d). Cdc45 accumulation peaked at 40 min at the *Ter* repeats but was also detected, to a lesser extent, at 30 and 50 min within the repeats (Fig. 1d). Using the same chromatin, we analyzed Tus binding dynamics to *Ter* array. As expected, HA-Tus binds with high affinity to the *Ter* repeats (Fig. 1e). Interestingly, we observed that the Cdc45 peak at *Ter* repeats coincides with the strongest reduction of Tus binding to *Ter* repeats (Fig. 1e). Therefore, we measured the DNA

copy number at the region covering the 9th and 10th *Ter* repeat. We observed DNA duplication of this region that is consistent with Cdc45 ChIP (Supplementary Fig. 1c). Therefore, the replicative helicase must have dislodged a few Tus proteins before stalling within the 9th and 10th *Ter* repeat. These findings are consistent with published data showing that Tus/Ter barrier causes an efficient replication fork stalling[41,42].

We next asked whether Set1C is recruited to replication forks (RF) stalled at the Tus/Ter barrier. We performed ChIP analysis of Myc-tagged Spp1, a subunit of Set1C[44], on the same chromatin samples that were used previously. Interestingly, we detected a Tus-dependent enrichment of Spp1 at the *Ter* repeats (Fig. 1f). Spp1 binding peaked at 40 min with minor detection at 30 min (Fig. 1f). These experiments suggest that Spp1 is recruited to the site of Tus/Ter dependent-stalled forks.

### Spp1 is recruited to the stalled fork independently of Set1C

Spp1 is a constitutive subunit of Set1C but was shown to have functions independent of its interaction with the complex during meiosis[44,45]. Henceforth, we wondered to which extent Spp1 detection at Tus/Ter barrier reflects Set1C occupancy. To this end, we performed ChIP-qPCR of Swd3, another Set1C subunit that associates with the SET domain of Set1, using the same pair of primers covering *Ter* region. While both Spp1 and Swd3 were enriched at highly transcribed *PMA1* gene, only Spp1 was detected at the stalled forks (Fig. 2a and Supplementary Fig. 2a). We further investigated the occupancy of Set1C subunits upstream of the *Ter* region (Fig. 2b). The Swd3 subunit was not detected within or around Tus/Ter barrier at the time of fork stalling. In contrast, Spp1 detection was not limited only to the *Ter* array but rather peaks transiently up to 0.8 kb upstream of the replication fork barrier (Fig. 2b and Supplementary Fig. 2b). These data suggest that Spp1 recruitment to chromatin as replication forks stall could be independent of its association with Set1C. To confirm this observation, we monitored RNA Polymerase II (RNA PolII) along the same regions since Set1C association with RNA PolII allows its recruitment to chromatin at transcribed regions[46–49]. We observed no significant accumulation of active RNA PolII during replication upstream Tus/Ter barrier (Fig. 2c). More importantly, we saw no overlapping between Spp1 and RNA PolII profiles at the time of RF stalling. Nonetheless, when comparing Spp1 and Cdc45 recruitment profiles, it becomes clear that Spp1 accumulates upstream of the stalled fork as Cdc45 detection maximizes at the *Ter* repeats (Supplementary Fig. 2c). Overall, these data demonstrate that Spp1 is recruited to chromatin upstream the Tus/Ter dependent perturbed replication fork, independently of its canonical pathway of recruitment.

### The parental chromatin mark serves as a target for Spp1 recruitment to stalled replication forks

Spp1 is not only important for H3K4me3 deposition but also has a PHD finger domain that enables its interaction with H3K4me3[44,45]. Spp1 acts independently of Set1C in meiosis by binding to H3K4me3[50,51]. Since Spp1 appears to be recruited upstream of the replication fork barrier independently of Set1C, we assessed whether Spp1 could be recruited via its PHD finger domain, especially because H3K4me3 is a surrogate mark for the parental histones[19,20]. To this end, we first monitored H3K4me3 density around Tus/Ter barrier and checked whether this mark would serve as a target for Spp1 binding as replication fork stalls. As previously done, H3K4me3, Cdc45-V5 and Spp1-Myc enrichments were quantified by ChIP-qPCR (Fig. 3a, b, c). As shown in Fig. 3b (see also Fig. 1d), the progressing replisome stalls within the *Ter* array at the time of 30 and 40 min in S phase. We found that H3K4me3 is present mainly in regions of 0.8 kb and 0.2 kb upstream of the Tus/Ter barrier, while this mark is not detected at ARS305 and at the *Ter* array, which is expected to exhibit minimal levels of H3K4me3 (Fig. 3b). The decrease

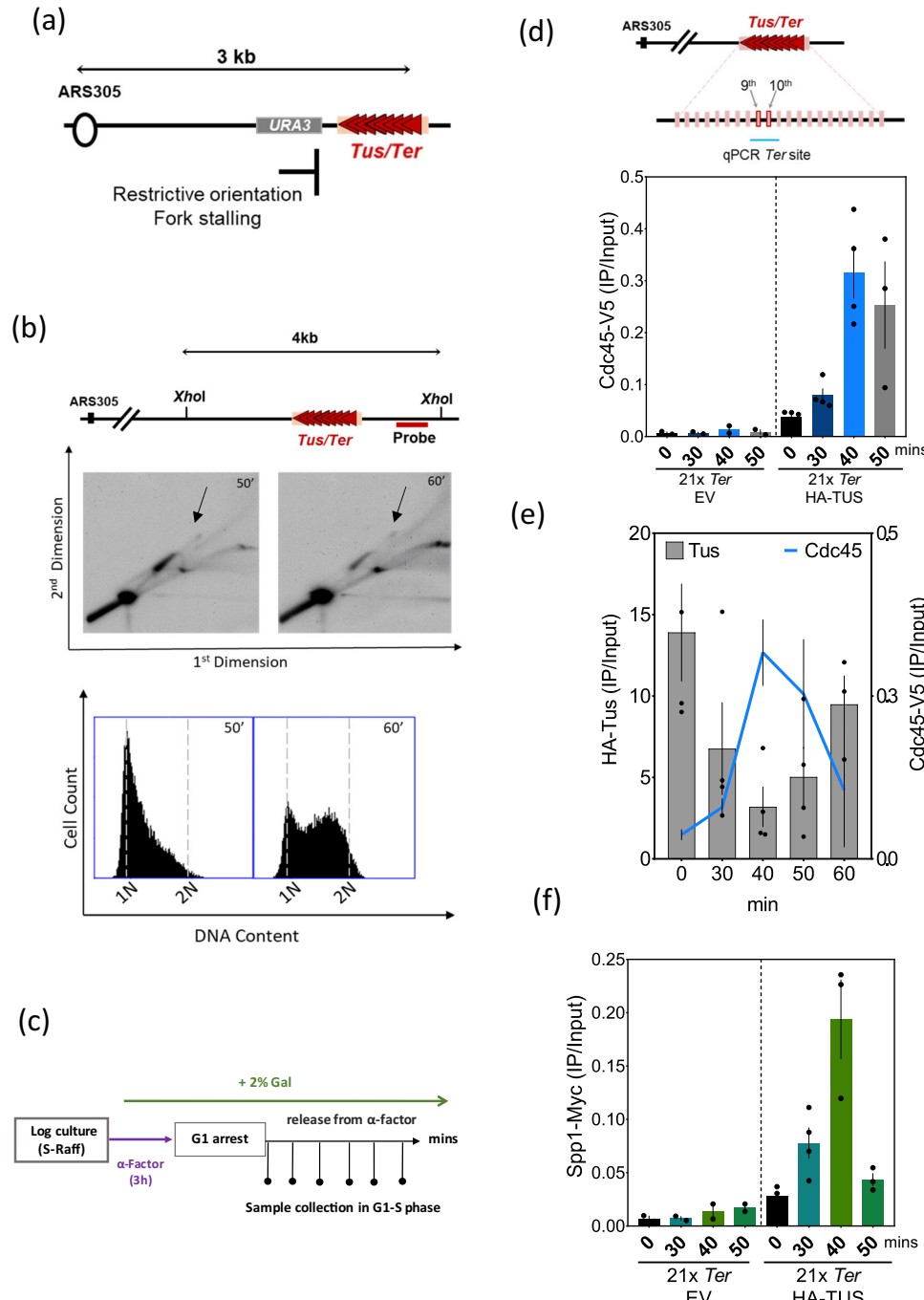

**Fig. 1 | Spp1 subunit of Set1C is recruited to stalled replication fork in early replication. a** Schematic representation of Tus/*Ter* barrier in the restrictive orientation relative to ARS305 on ChrIII. Tus binds specifically to the *Ter* sequence inducing replication fork stalling. **b** Log culture, grown in S-raffinose at 25 °C, was arrested in G1 by the addition of 3 μg/ml of α-factor for 3 h and, after 30 min, 2% galactose was added to induce Tus expression. Samples were collected at 50 and 60 min after G1 arrest release. Genomic DNA was cut with Xho*I* restriction enzyme whose cut sites flank the barrier region producing a 4 kb fragment. Subsequently, it is visualized using the indicated specific probe. 2D agarose gel analysis of the Tus/*Ter* dependent-stalled forks at 50 and 60 min are shown. The "tear" shaped spot represents the paused forks and diminishes at 60 min while the arrows show the X-shaped DNA structures. Bottom, FACS profiles of the corresponding 2D gel samples. **c** Outline of experimental procedures used to collect samples that are used in the subsequent experiments. Cells were grown in S-Raffinose and blocked in G1 for 3 h; after 30 min addition of α-factor, 2% galactose was then added to

induce HA-Tus expression. Cells were released into the S phase in pre-warmed SD-GAL media. The G1 time point corresponds to T = 0 collected samples immediately after washing. **d** Top; Diagram showing the *Ter* pair of primers covering the region between 9th and 10th *Ter* repeats used for the qPCR. Bottom: ChIP-qPCR analyses of Cdc45-V5 at different time points after release from G1 arrest in the strain expressing either HA-Tus or an empty vector (negative control). Data are represented as mean value ± SEM and correspond to *n* = 4 independent experiments for 21x*Ter* HA-TUS and *n* = 2 independent experiments for 21x*Ter* EV. **e** ChIP-qPCR profile of Cdc45-V5 plotted against ChIP-qPCR profile of HA-Tus, using the same chromatin as Cdc45 ChIP. SEM represents four independent experiments. **f** ChIP-qPCR profile of Spp1-Myc in strain either without or with Tus expression, using the same chromatin as previous Cdc45 ChIP and the same pair of primers. Data are represented as mean value ± SEM and correspond to *n* = 4 independent experiments for 21x*Ter* HA-TUS and *n* = 2 independent experiments for 21x*Ter* EV. Source data are provided as a Source Data file.

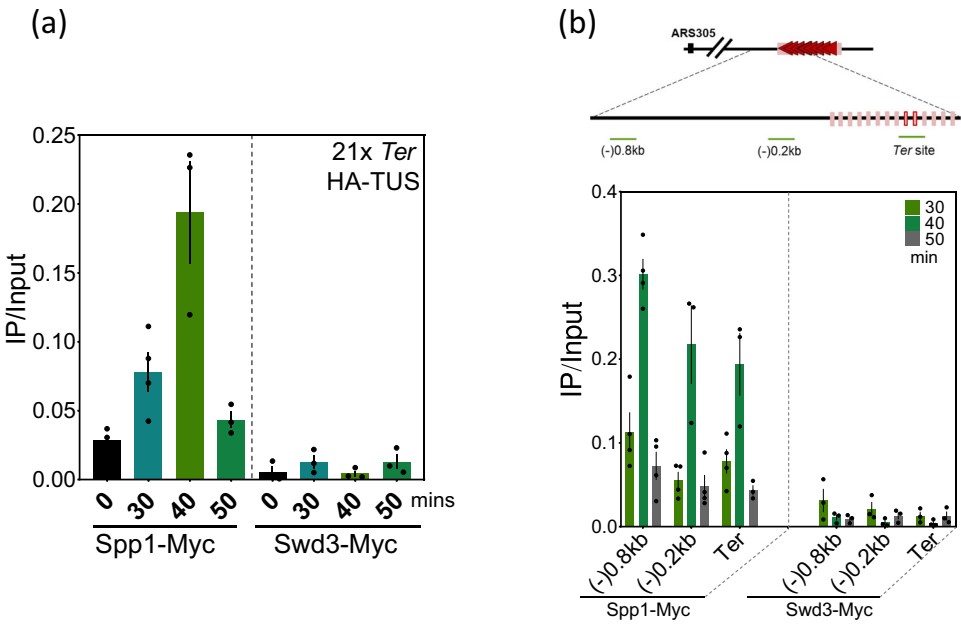

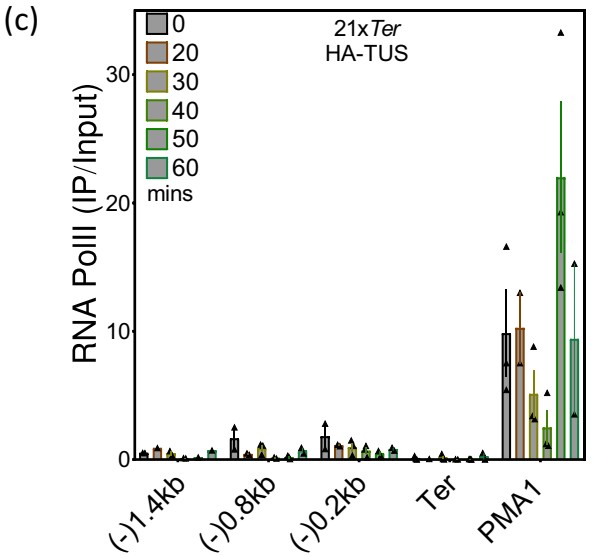

**Fig. 2 | Spp1 recruitment to stalled fork is independent of Set1C. a** ChIP-qPCR of Swd3-Myc and Spp1-Myc at *Ter* site in cells expressing Tus. The samples were collected as described in Fig. 1c. Data are represented as mean value ± SEM and correspond to $n = 4$ independent experiments for Spp1-Myc and $n = 3$ independent experiments for Swd3-Myc. **b** Top, schematic representation of the pair of primers used upstream of the Tus/*Ter* barrier used for ChIP-qPCR. Bottom, ChIP-qPCR analyses of Swd3-Myc and Spp1-Myc at the indicated regions at 30, 40, and 50 min after release from α-factor. Data are represented as mean value ± SEM and correspond to $n = 4$ independent experiments for Spp1-Myc and $n = 3$ independent experiments for Swd3-Myc. **c** ChIP-qPCR of RNA PolII at regions within and around *Ter* array. *PMA1* (a highly transcribed gene) is used as a positive control of the ChIP experiment. Data are represented as mean value ± SEM and correspond to $n = 3$ independent experiments for Spp1-Myc and $n = 2$ independent experiments for Swd3-Myc. Source data are provided as a Source Data file.

in H3K4me3 at these regions likely reflects dilution of parental chromatin marks after replication (Supplementary Fig. 3a)[18,20].

Using the same chromatin samples, we detected Spp1 enrichment upstream of the stalled fork, which overlaps with the H3K4me3 profile (Fig. 3c). These data suggest that H3K4me3 serves as a binding site for Spp1 recruitment upon replication fork stalling. We reasoned that deleting the PHD finger domain of Spp1 should abolish its recruitment. To test this hypothesis, we built a strain where Spp1 is devoid of its PHD domain (spp1$^{\Delta PHD}$) and compared Spp1 levels on chromatin between WT (Spp1-Myc) and Spp1-Myc$^{\Delta PHD}$. In agreement with our hypothesis,

we observed that Spp1 occupancy at the stalled fork was abolished in the spp1$^{\Delta PHD}$ strain (Fig. 3d). It is worth noting that Spp1 loading at active genes (i.e., *RPL2a, PMA1*) was decreased in spp1$^{\Delta PHD}$ strain (Supplementary Fig. 3b) indicating that Spp1 binding to H3K4me3 may help Set1C recruitment to chromatin[51]. These data demonstrate that H3K4me3 recognition by the PHD domain is important for Spp1 recruitment to Tus/*Ter*-dependent-stalled forks. Noteworthy, these findings are supported by recent data showing that at weakly transcribed genes, Spp1 binds to parental histones (H3K4me3) via its PHD finger domain following DNA replication[52]. Nonetheless, our data

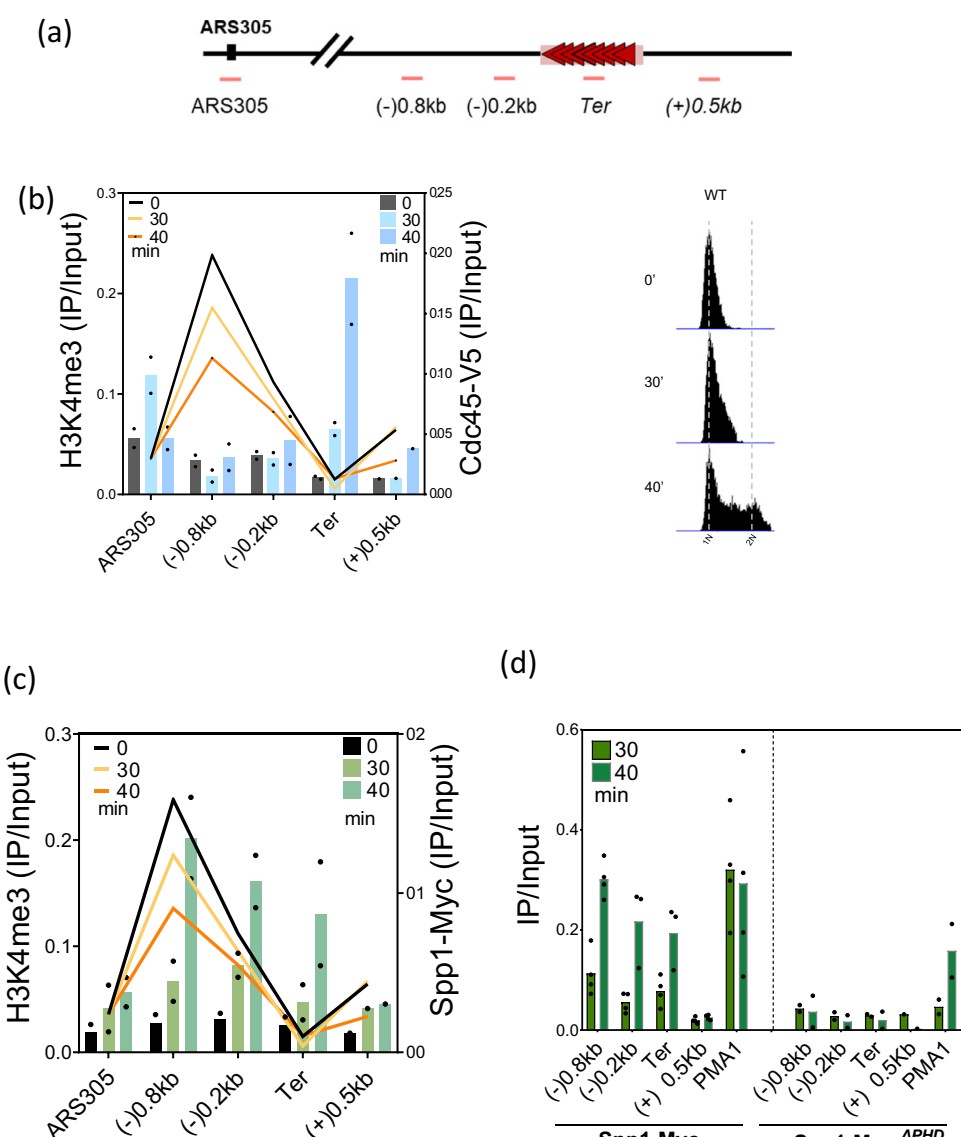

**Fig. 3 | Spp1 and H3K4me3 occupancy at regions surrounding the Tus/*Ter* barrier. a** Schematic representation of the pair of primers used for the ChIP-qPCR at regions surrounding and within the *Ter* array. **b** Right, cell cycle progression analyzed by FACS. Cells were synchronized by adding three subsequent reduced doses of α-factor to increase the proportion of cells in the early S phase at 30 min. Left, overlapping ChIP-qPCR profile of H3K4me3 (IP/Input) and Cdc45-V5 (IP/Input) at 0, 30, and 40 min after release from G1 arrest. Both ChIP experiments were done on the same chromatin. Data are represented as mean value ± SEM of *n* = 2 independent biological replicas of H3K4me3 colored line graph and of Cdc45 in a blue-colored bar graph. **c** Overlapping ChIP-qPCR profile of H3K4me3 and Spp1-Myc. Similarly to (**b**), H3K4me3 is represented by an orange line graph, while the Spp1 profile is represented by the green bar graph. The ChIP experiments were done from the same chromatin samples as those of (**b**). Data are presented as mean value ± SEM of *n* = 2 independent experiments. **d** ChIP-qPCR of Spp1-Myc and in Spp1-Myc^*ΔPHD* at 30 and 40 min in S phase. *PMA1* is used as a positive control. Growth conditions and sample collection was done as shown in (**c**). Data are presented as mean value ± SEM of *n* = 2 independent experiments. Source data are provided as a Source Data file.

overall suggest that Spp1 recruitment to the nascent chromatin behind Tus/Ter-dependent stalled forks is dependent on the PHD finger domain.

## The dynamics of Tus/*Ter*-dependent replication fork stalling is altered in spp1 mutants

To understand the role of Spp1 at stalled forks, we analyzed the consequences of deleting either the full-length SPP1 or its PHD finger domain. As previously done, we monitored replisome progression by Cdc45 ChIP in spp1Δ and spp1^ΔPHD strains. While in WT cells, Cdc45 was mainly detected within the *Ter* repeats at 40 min, in both spp1 mutants, Cdc45 was detected at 30 min, indicating an earlier replication fork stalling at the Tus/*Ter* barrier in both spp1 mutants (Fig. 4a, left).

In parallel, we analyzed the global cell cycle progression by FACS under the same growth conditions used in previous experiments. We noticed an acceleration in bulk DNA synthesis in both spp1 mutants (Fig. 4a, right), which could explain why replication fork stalling is observed at earlier time points in these mutants. We next monitored the dynamics of the Tus/*Ter* dependent stalled fork by 2D gels at 30 and 40 min after release from α-factor (Fig. 4b). Interestingly, we detected a strong replication fork stalling in spp1Δ at 30 min, but the signal disappeared at 40 min (Fig. 4b; top, right). At the same time points, the stalled replication fork signal is weaker in WT cells at 30 min but remains visible until 40 min in S phase (Fig. 4b; top, left). Of note, the FACS profile of WT and spp1Δ (Fig. 4b; bottom) shows that both strains, under these experimental conditions, had a comparable number of cells in the early S phase. These data suggest that in the

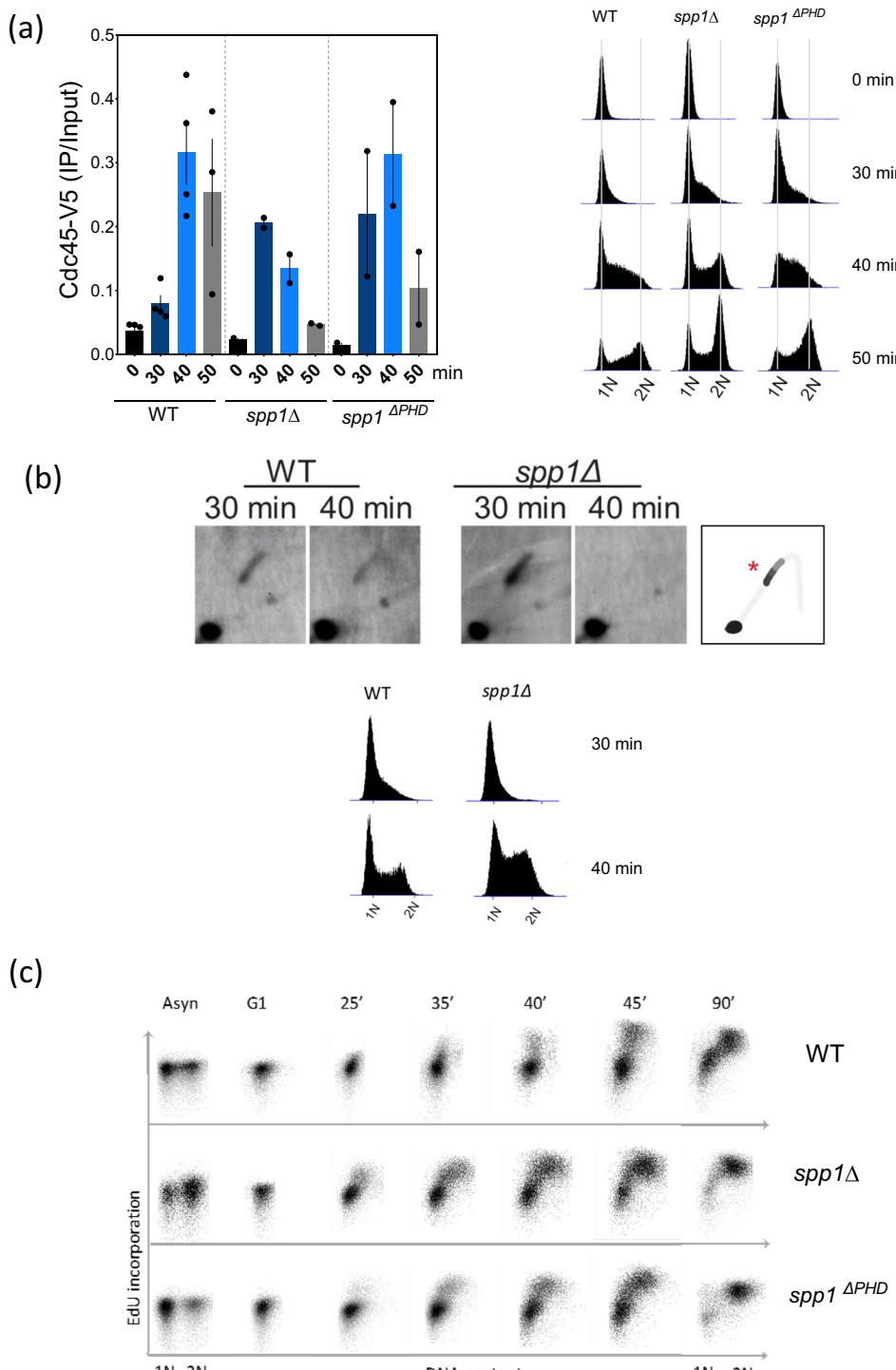

**Fig. 4 | Replication fork progression is altered in spp1 mutants. a** Left, ChIP-qPCR within the *Ter* array of Cdc45-V5 in WT, spp1Δ and spp1*ΔPHD* strains. The same conditions were used for the three strains. Cells were synchronized for 3 h in alpha factor and released in a pre-warmed S-Raffinose (+GAL); samples were collected at 0, 30, 40, and 50 min. Data are presented as mean values of ± SEM of *n* = 4 independent experiments in WT and *n* = 2 independent experiments in spp1Δ and spp1*ΔPHD*. Right, representative FACS profile of the cell cycle progression of WT, spp1Δ and spp1*ΔPHD* strains corresponding to the Cdc45 ChIP. **b** Log cultures, grown in S-raffinose at 30 °C, were arrested in G1 by the addition of 3 µg/ml of α-factor for 3 h and, after 30 min, 2% galactose was added to induce Tus expression. Top, genomic DNA was cut with Xho*I* restriction enzyme and visualized using the same probe as in Fig. 1. 2D agarose gel analysis of the Tus/*Ter* dependent-stalled forks in WT and spp1Δ at 30 and 50 min are shown. The "tear" shaped spot represents the paused forks also represented by the red star in the schematic drawing (right). Bottom, FACS profiles of the corresponding 2D gel samples. **c** FACS profiles with bivariate EdU Alexa 647 vs propidium iodide (PI). The vertical shift reflects EdU-incorporated DNA, subsequently representing cells undergoing replication in S phase, while propidium iodide (PI) reflects DNA content. Cells were synchronized in G1 as previously described, but 5 µg/ml alpha factor was added instead of 10 µg/ml to enhance cell release to S phase. Cells were then released to S phase in media containing 25 µM EdU for 20 min and chased by 10X thymidine. Meanwhile, cells were collected at different time points for click reaction and FACS analysis. The experiments were done independently three times. Source data are provided as a Source Data file.

spp1Δ mutant, replication fork stalling occurs earlier and is stronger. However, we cannot rule out the possibility that this phenotype is in part due to a higher fraction of early S-phase cells in the spp1Δ mutants. Therefore, we used WT, spp1Δ, and spp1^ΔPHD strains capable of incorporating the thymidine analog (EdU) during DNA synthesis to track cell cycle progression with spatial and temporal resolution[53]. The percentage of cells in S phase was determined by FACS analysis of incorporated EdU. The results indicate that the percentage of cells that have incorporated EdU is higher at the beginning of S phase in both spp1 mutants. Strikingly, we observed a faster progression through S phase of spp1Δ cells and, to a lesser extent, of spp1^ΔPHD cells (Fig. 4c).

To assess whether the replisome moves faster in spp1Δ cells, we monitored replication fork progression by molecular combing of DNA fibers from asynchronous WT and spp1Δ cells (Supplementary Fig. 4). Compared to WT, the spp1Δ mutant has longer tracks of newly replicated DNA, reinforcing the idea that replication forks progress more rapidly in the absence of Spp1 in S-Raffinose.

We conclude that the absence of Spp1 affects the timing and strength of the Tus/Ter-dependent replication fork stalling.

### Spp1 restricts ssDNA formation at Tus/Ter barrier

It is known that fork stalling at Tus/Ter barrier causes Exo1-dependent ssDNA gap formation upstream of the stalled forks[42,43]. Therefore, we monitored RPA occupancy at regions surrounding and at the Ter sequence using the same chromatin samples of the previous experiments (Fig. 5a and Supplementary Fig. 5a). In WT cells, RPA was detected at the regions of (−) 0.8 kb and (−) 0.2 kb at 30 and 40 min, respectively (Fig. 5a and Supplementary Fig. 5a). We postulated that we first detected fork-associated RPA-ssDNA and we further detected ssDNA generated by fork processing upstream the stalled forks (Fig. 5b), an interpretation in agreement with previous studies[42,43].

To determine whether Spp1 regulates RPA accumulation at stalled forks, we measured RPA levels upstream of the Tus/Ter barrier in spp1Δ and spp1^ΔPHD cells. As in WT cells, RPA-coated ssDNA profiles behind the stalled fork in both spp1 mutants are consistent with the profiles of Cdc45 occupancy (Fig. 5c and Supplementary Fig. 5b, c). While RPA levels peak at the 9th and 10th Ter repeat at 40 min in WT cells, in spp1Δ and spp1^ΔPHD mutants, RPA binding dramatically increases and occurs earlier (at 30–40 min) at the stalled forks (Fig. 5d and Supplementary Fig. 5d). This is likely due to the earlier replication fork stalling observed in spp1 mutants. Since RPA binding reflects ssDNA formation, these data suggest that Spp1 and its PHD finger domain are important to prevent ssDNA accumulation at sites of replication fork stalling.

As ssDNA formation at Tus/Ter-dependent stalled fork is caused by Exo1 activity[42], we sought to monitor RPA level in spp1Δ exo1Δ double mutant. Interestingly, RPA occupancy was diminished not only upstream of the stalled fork but also within Ter repeats in the double mutant (Fig. 5e). Thus, the ssDNA accumulation in the absence of Spp1 was dependent on Exo1 activity. Therefore, we conclude that Spp1 restricts nucleolytic degradation of nascent DNA through a process requiring its PHD domain.

### The absence of Spp1 increases mutagenesis upstream of the Tus/Ter barrier

Our data suggest that Spp1 loss increases ssDNA formation at Tus/Ter dependent-stalled forks. Since ssDNA is extremely prone to hypermutation[54], we quantified the mutation rate as fork stalls at the Tus/Ter barrier in the presence or absence of Spp1. We took advantage of the URA3 reporter gene located immediately upstream of the Tus/Ter barrier to quantify the mutation rate as previously described[42]. In WT cells, expression of the Tus protein slightly increased the URA3 mutation rate (Fig. 6, top), most mutations being either substitutions or small insertions (Fig. 6 bottom, left). In spp1Δ cells, we observed a significant increase in mutation rate that was

dependent on Tus expression (Fig. 6, top, right). Interestingly, microdeletions (<3 bp) were increased by tenfold in spp1Δ cells. Because the increased mutation rate is one of the characteristics of increased ssDNA formation[62,63], these results are consistent with Spp1 causing excessive ssDNA formation when the replication fork stalls. We next tested whether the increase in URA3 mutation rate was dependent on HR. We therefore measured the URA3 mutation rate in rad52Δ and rad52Δ spp1Δ cells, RAD52 being essential for HR in S. cerevisiae. We chose to delete RAD52 rather than RAD51 because analysis of the URA3 mutation rate in rad51Δ cells is complicated by the fact that deleting RAD51 greatly increases the mutagenic rate, even in the absence of Tus[42]. Interestingly, we observed that both deletions of rad52Δ and spp1Δ increased the URA3 mutation rate to the same extent in cells expressing the Tus protein (Fig. 6, top). In rad52Δ cells, mutations were found to be exclusively microdeletions (Fig. 6, bottom). Interestingly, deleting both SPP1 and RAD52 has additive effects on the URA3 mutation rate. However, in the double rad52Δ spp1Δ mutant, most of the repair events were found to be microdeletions and microinsertions, as was the case for the single spp1Δ mutant (see "Discussion").

### Spp1 restricts ssDNA formation at the Tus/Ter barrier by protecting nascent chromatin during fork stalling

The mechanism by which Spp1 restricts ssDNA formation remains unclear. We sought to investigate the nascent chromatin organization during Tus/Ter-dependent fork stalling by verifying whether its protection against MNase digestion is altered in the absence of SPP1. To this purpose, cells were arrested in G1 in an S-Raff (+GAL) medium and released from the G1 arrest in the presence of EdU to allow its incorporation into nascent DNA; cells were then collected at 20, 30 and 40 min (Supplementary Fig. 6). Samples were split into three parts and analyzed for histone H3 ChIP, undigested chromatin and MNase-digested chromatin (Fig. 7a). We next employed the click reaction to conjugate biotin with EdU-labeled DNA enabling nascent DNA recovery by streptavidin pulldown[55, 56] (Fig. 7a). We performed streptavidin pulldown on both MNase-digested and sonicated (undigested) DNA to allow quantification of the protected nascent chromatin. Recovered nascent DNA was then analyzed by qPCR using the same pair of primers used in the previous experiments, excluding the sub-nucleosomes fragments.

We first compared nucleosome occupancy by monitoring the histone density (H3 ChIP) at regions surrounding Tus/Ter barrier (Fig. 7b). We observed a slight decrease in H3 density in WT and spp1Δ cells at 30 min, especially in the proximal regions of the barrier (Fig. 7b). Interestingly, the H3 density was rather restored at 40 min, which might reflect the passage of the replication fork along the chromatin. We also observed poor chromatin protection at ARS305 in WT and spp1Δ cells, which is expected because the origins of replication are depleted nucleosome regions and thus highly susceptible to MNase digestion (Fig. 7c, left). In WT cells, chromatin protection decreased in regions upstream of the Ter site at 30 and 40 min (compared to 20 min), reflecting chromatin remodeling during replication fork stalling at the Tus/Ter barrier (Fig. 7c, left). Interestingly, we observed a dramatic decrease in chromatin protection in the absence of Spp1 (Fig. 7c; right). Noteworthy, at 1.4 kb upstream of the Tus/Ter barrier, we detected reduced chromatin protection in both WT and spp1Δ cells. However, at 40 min, chromatin organization was restored in spp1Δ cells but not in WT cells. As discussed above, we believe that this is due to the difference in fork stalling dynamics. Nevertheless, the strong increase of chromatin accessibility observed in spp1Δ cells reflects an increase in chromatin remodeling as the replication fork stalls at Tus/Ter barrier. Higher nascent DNA accessibility in the absence of Spp1 could be due to the increased activity of fork remodelers, subsequently creating an entry point for nucleases in the absence of protective factors[10,12,14,57].

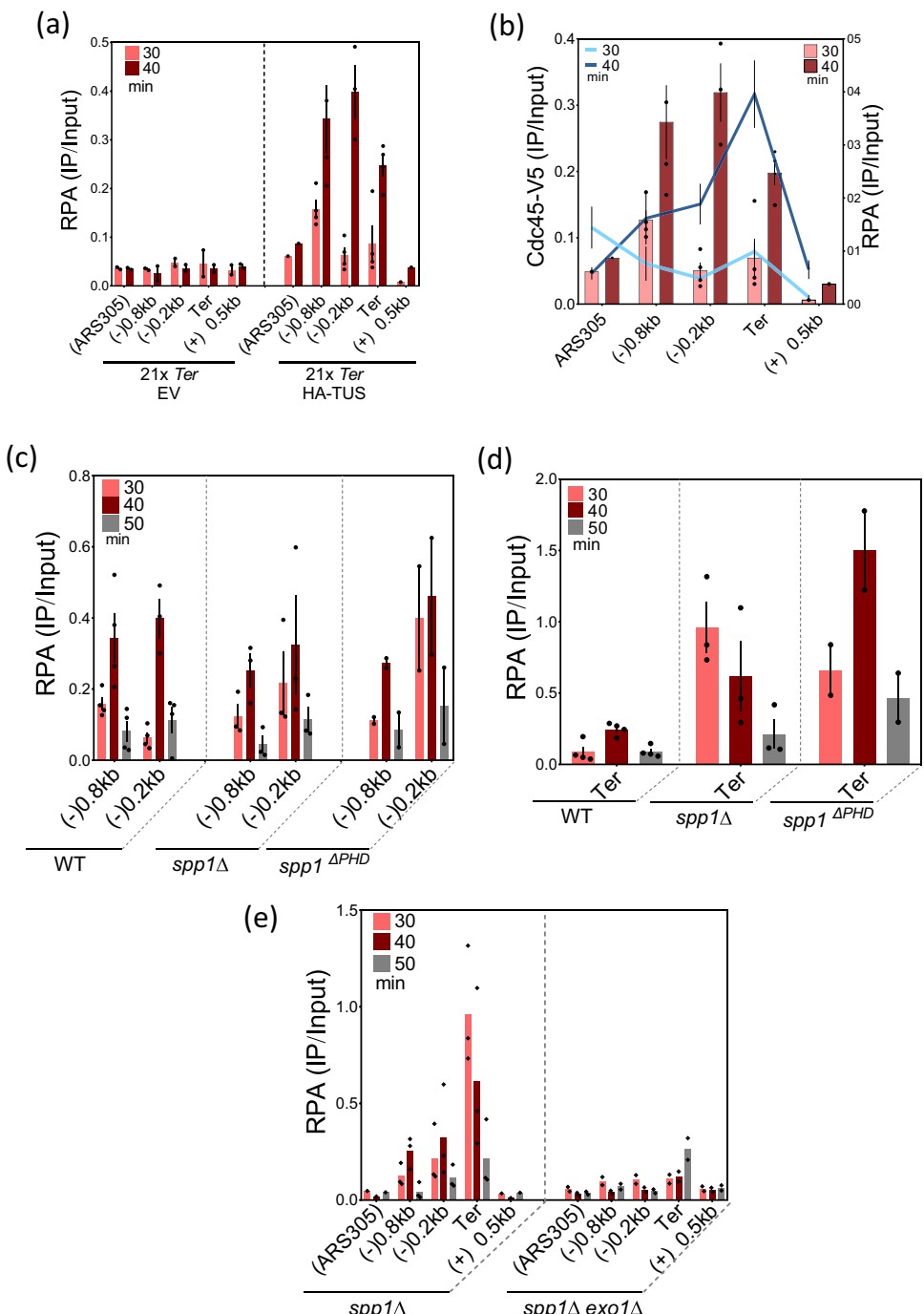

**Fig. 5 | RPA occupancy at Tus/*Ter* stalled fork in WT, spp1Δ and spp1$^{ΔPHD}$ strains. a** ChIP-qPCR of RPA in EV and HA-Tus expressing strain. Where RPA occupancy is measured in the indicated regions. Data are represented as mean value ± SEM and correspond to *n* = 4 biologically independent experiments for 21xTer HA-TUS and *n* = 2 biologically independent experiments for 21xTer EV. **b** Overlapping occupancy of RPA and Cdc45 at Tus/*Ter* and flanking regions. Only times 30 and 40 min are shown for clarity and simplicity of the figure. The red bar graphs and the blue lines represent RPA and Cdc45 levels, respectively. Data are represented as mean value ± SEM and correspond to *n* = 4 biologically independent experiments. **c** Comparison of RPA occupancy (ChIP-qPCR) between WT, spp1Δ

and spp1$^{ΔPHD}$ at region upstream of the barrier. Data are represented as mean value ± SEM and correspond to *n* = 4 of biologically independent experiments for WT, and *n* = 3 of biologically independent experiments for spp1Δ and *n* = 2 for spp1$^{ΔPHD}$. **d** Same as (**c**), but RPA occupancy is measured in the *Ter* region. All RPA levels were quantified from the same chromatin used to monitor Cdc45 occupancy. **e** Comparison of RPA ChIP-qPCR profiles between spp1Δ and spp1Δ exo1Δ at regions surrounding the Tus/*Ter* replication fork barrier. Data are represented as mean value ± SEM and correspond to *n* = 3 biologically independent experiments in spp1Δ and *n* = 2 biologically independent experiments *in* spp1Δ exo1Δ. Source data are provided as a Source Data file.

Taken together, these results indicate that Spp1 binding to chromatin is important for nascent chromatin protection during replication fork stalling at the Tus/*Ter* barrier.

**Deleting SPP1 sensitizes cells to RPA dysfunction**

Our results indicate an increase of RPA-ssDNA at the Tus/*Ter* barrier in the spp*1* mutants. We thus wondered whether the absence of Spp1

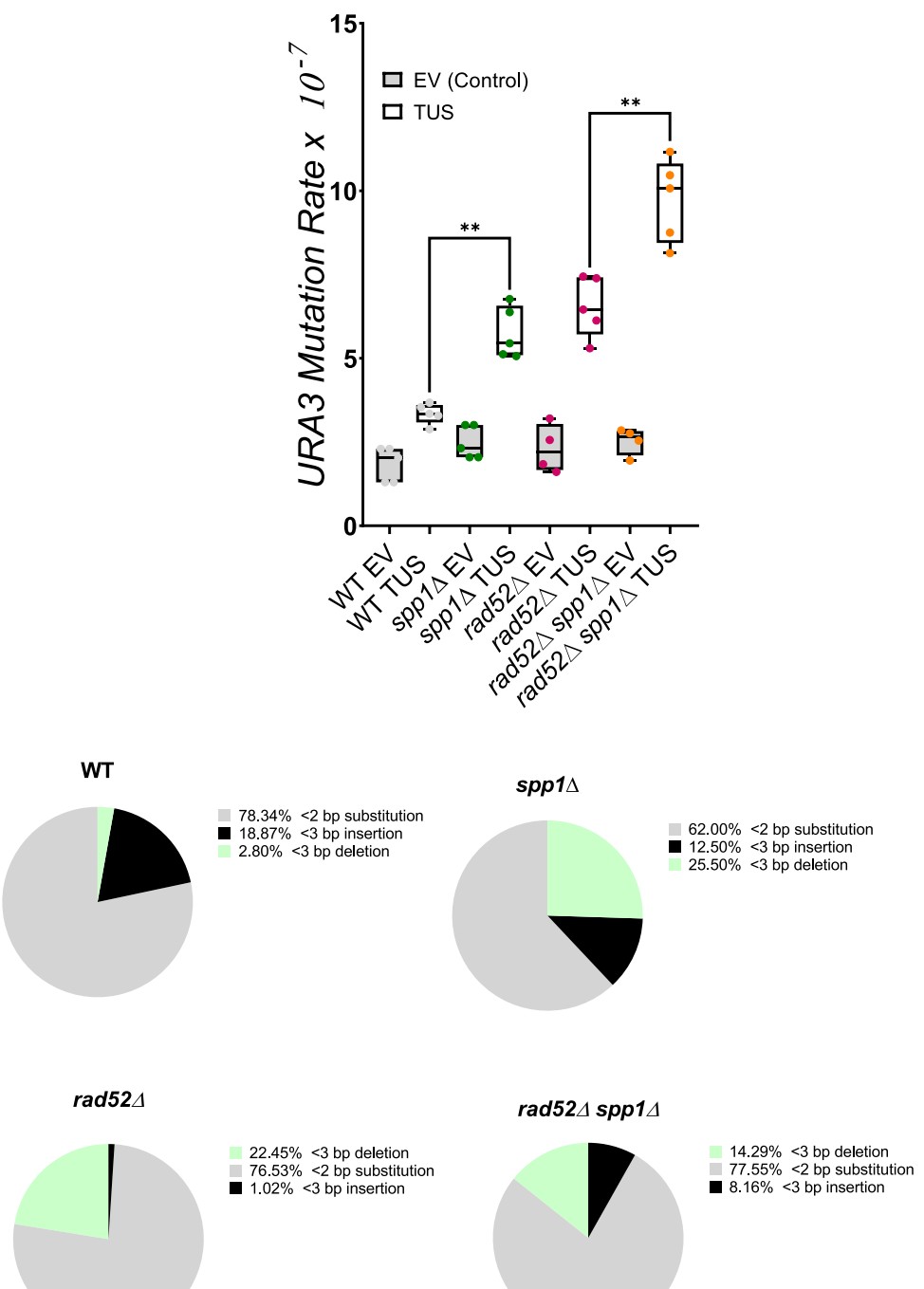

**Fig. 6 | Tus/*Ter* proximal *URA3* mutagenesis in the absence of Spp1.** Serial dilution of exponentially growing cells (in galactose) was plated out on 5-FOA and then selected for *URA3* mutations. For each strain, over 100 colonies were counted. Top, Box-and-whisker plots with the upper and lower quartile with a median show the mutation rate in WT, spp1Δ, *rad52Δ* and *rad52Δ* spp1Δ strains with or without Tus protein expression. Statistical analyses were done (*n* = 5 biologically independent experiments for all except for *rad52Δ* EV and *rad52Δ* spp1Δ EV; *n* = 4) with a two-tailed Mann–Whitney test; * *p* < 0.05; ** *p* < 0.005; **** *p* < 0.0001; ns, not significant. Bottom, *URA3 was* sequenced. The types of *URA3* mutations in WT, spp1Δ, *rad52Δ* and *rad52Δ* spp1Δ (expressing Tus) are shown in the pie charts. Source data are provided as a Source Data file.

could sensitize cells to global RPA dysfunction. To this end, we first quantified the percentage of RPA (Rfa1-CFP) and Rad52 (Rad52-YFP) foci in S-Raff (±Camptothecin). We choose to use the topoisomerase 1 inhibitor camptothecin (CPT) because it induces fork reversal which has been proposed to stabilize replication forks[58]. Surprisingly, RPA and Rad52 foci increased in spp1Δ cells compared to WT cells in S-Raff, even without CPT (Fig. 8a). Nevertheless, the addition of CPT further increased the percentage of Rfa1 foci in WT and spp1Δ cells, and to a

lower extent the one of Rad52 foci. Of note, the RPA foci observed in spp1Δ cells are brighter, likely reflecting longer stretches of RPA-bound ssDNA. These data indicate that the absence of Spp1 leads to a global increase in ssDNA formation (Fig. 8a).

To further validate that the lack of Spp1 sensitizes cells to RPA dysfunction, we deleted *RTT105*, an RPA chaperone that regulates RPA levels at the ssDNA[59,60]. Our genetic analyses show that the inactivation of Spp1 in *rtt105Δ* cells has little effect on growth but results in a strong

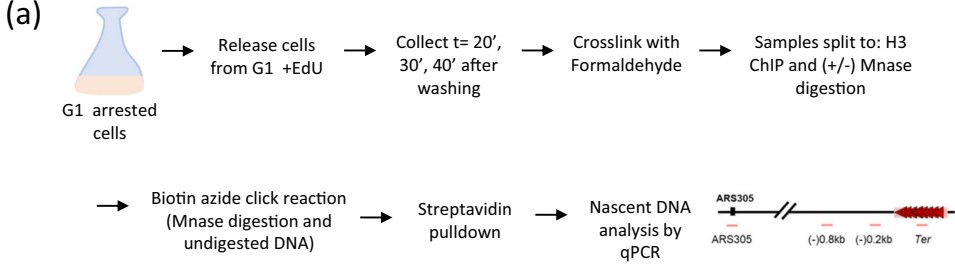

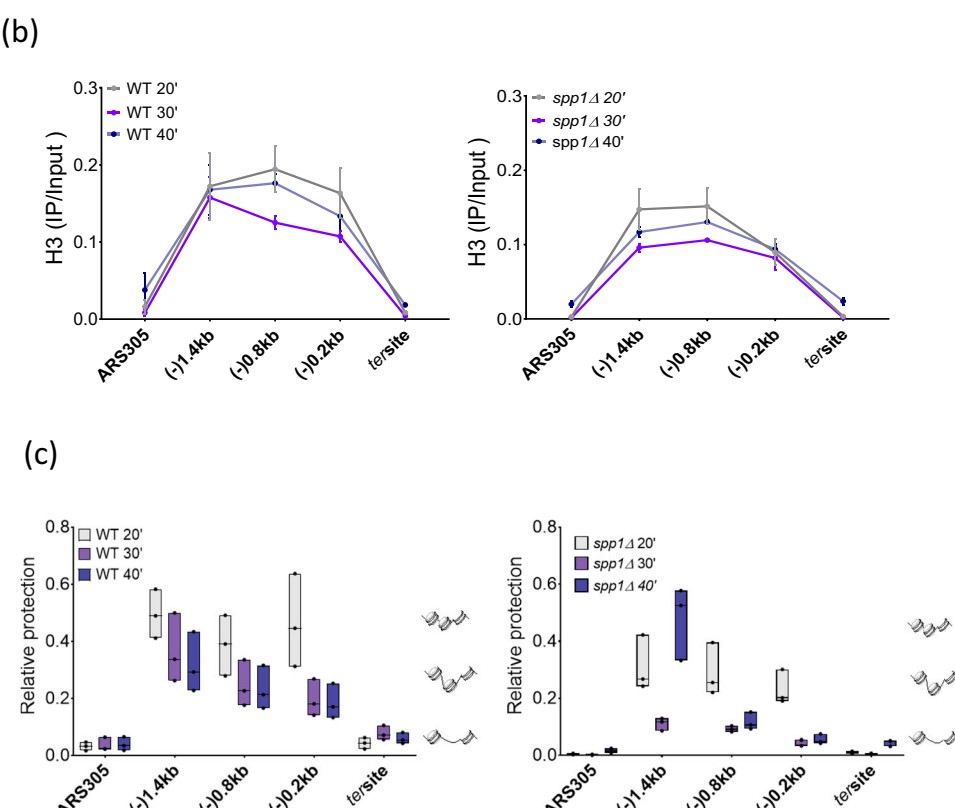

**Fig. 7 | Spp1 protects nascent DNA at forks stalled at the Tus/*Ter* barrier.**
**a** Scheme of the experiment. Histone ChIP and chromatin MNase digestion (or sonication of undigested chromatin) were performed from the same samples divided into three. Biotin azide click reaction and streptavidin pulldown were performed on both MNase-digested and undigested sonicated chromatin to isolate the nascent DNA. **b** Profiles of H3 ChIP (IP/Input) at 20-, 30- and 40-min after release from α-factor in WT and spp1Δ in cells expressing Tus protein. The qPCR was done using a pair of primers covering the regions between ARS305 and *Ter* array. Data are

represented as mean value ± SEM and correspond to *n* = 3 biologically independent experiments. **c** Floating bars (line at median) show the relative nascent DNA protection between WT and spp1Δ at 20-, 30- and 40-min after release from α-factor. Data corresponds to *n* = 3 biologically independent experiments. Relative protection is obtained by normalizing the MNase-qPCR of nascent DNA to undigested-qPCR of sonicated nascent DNA. Regions used for qPCR are represented. Source data are provided as a Source Data file.

increase in CPT sensitivity (Fig. 8b). Furthermore because we found that Exo1 was primarily responsible for the formation of RPA-bound ssDNA at the Tus/*Ter* barrier, we investigated whether deletion of *EXO1* would suppress the increased sensitivity to CPT in *rtt105Δ* spp1Δ cells. We found that deleting *EXO1* did not rescue the CPT sensitivity (Supplementary Fig. 7). This can be explained by the fact that other nucleases could be responsible for the global ssDNA formation in the absence of Spp1.

We next sought to deplete RPA in a cell cycle-dependent manner. Hence, we generated strains having *RFA1* under either CLB2 or CLB6 promoters that are repressed in S phase or G2/M, respectively (Fig. 8c). This system severely depletes RPA but retains residual RPA levels, avoiding cell lethality[60]. The *pCLB2-RFA1* itself was sensitive to CPT because RPA is needed during S phase when

replication forks are challenged. Interestingly, we observed an increase of CPT sensitivity in spp1Δ cells when RPA was depleted in S phase but not in G2/M (Fig. 8c). These data strengthen the notion that RPA becomes critical to protect excessive ssDNA that accumulates at CPT-induced lesions in spp1Δ cells. In addition, we assessed spp1Δ sensitivity to different genotoxic stresses (CPT, HU, MMS) in combination with the *rfa1-D228Y* and *rfa1-t11* mutants. While the *rfa1-D228Y* mutant decreases RPA affinity to ssDNA[61] and affects general RPA functions[62], the *rfa1-t11* mutant is defective in HR and fails to stabilize the stalled replication fork[63–65]. Loss of Spp1 itself does not increase sensitivity to CPT, HU, MMS at the concentrations tested. However, we observed a clear increase of drug sensitivity in *rfa1-D228Y* spp1Δ *and rfa1-T11* spp1Δ compared to the single *rfa1* mutants (Fig. 8d), further indicating that a fully

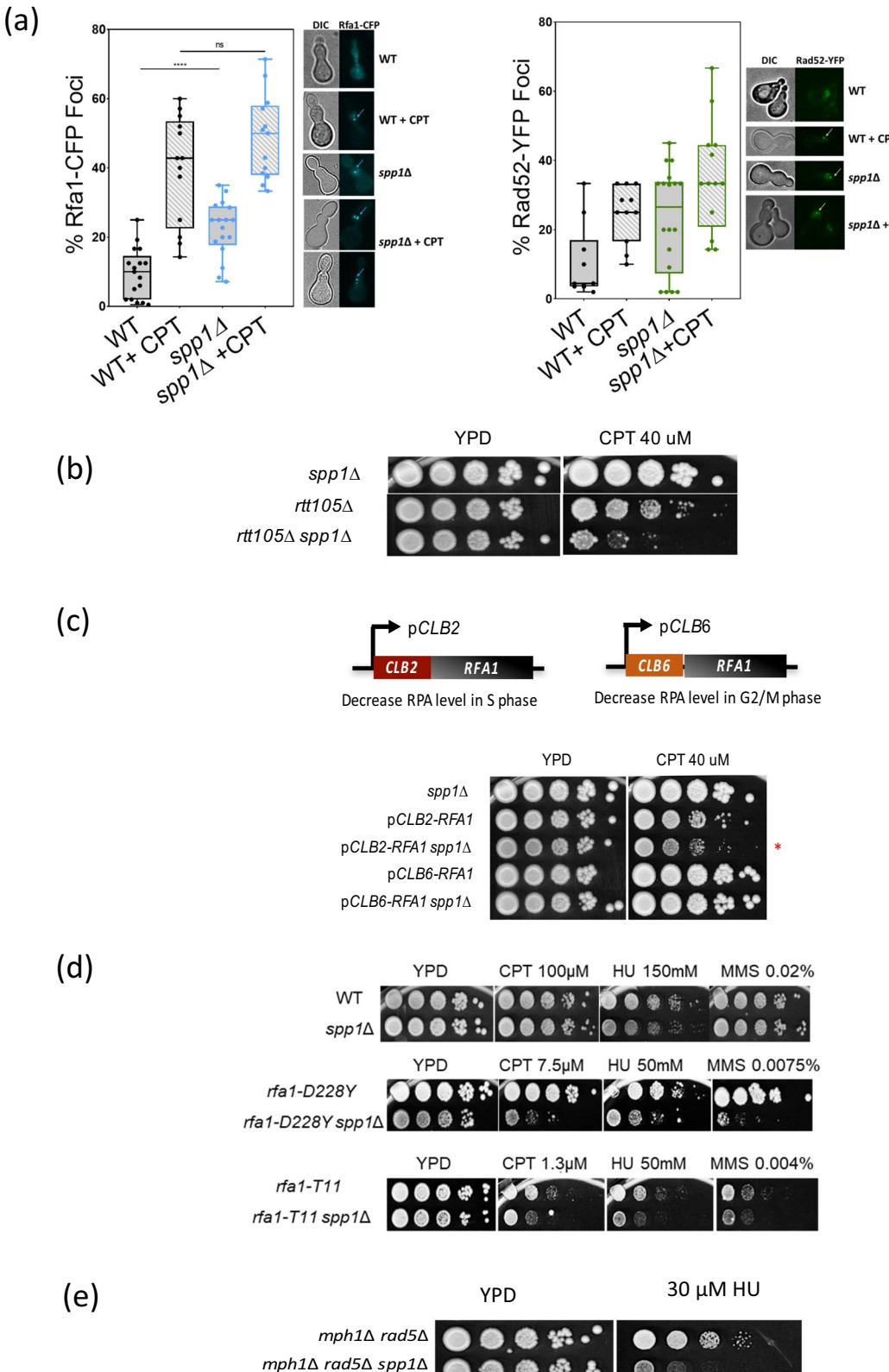

functional RPA is required in cells lacking Spp1 exposed to different replication stress-inducing agents.

Finally, it has been shown that Mph1/Rad5 promotes fork reversal at Tus/*Ter*-dependent replication fork stalling[43]. We thus asked whether the *mph1Δ rad5Δ* double mutant would manifest sensitivity to moderate HU concentrations upon SPP*1* deletion (Fig. 8e). The triple *mph1Δ rad5Δ* spp1Δ was sensitive to a low dose of HU, suggesting that

fork reversion becomes necessary in this setting, at the cost of increased nascent DNA degradation.

## Discussion

DNA replication occurs in the context of chromatin, where the replisome might encounter obstacles leading to replication fork perturbation. While the role of chromatin in response to DSB is established, it

**Fig. 8 | The absence of Spp1 sensitizes cells to RPA dysfunction. a** Box-and-whisker plots show the percentage of Rfa1-CFP (left) and Rad52-YFP (right) foci observed in WT and spp1Δ cells grown in S-Raffinose (±CPT, 50 μM) 40 min after release from G1 arrest. Statistical analysis is performed using a two-tailed Mann–Whitney test from n = 100 cells examined over three independent experiments, n.s., not significant. **p < 0.01, ****p < 0.0001. Examples of Rfa1-CFP and Rad52-YFP foci are shown. Examples of foci are indicated by the arrow. **b** Tenfold serial dilutions of exponentially growing cells with the indicated mutations were spotted onto YPD+CPT (40 μM) plates and incubated at 30 °C for 3 days. **c** Top, schematic representation of the *CLB2/CLB6-rfa1* constructs used in this study. Bottom, tenfold serial dilutions of exponentially growing cells with the indicated mutations were spotted onto YPD+CPT 40 μM plates and incubated at 30 °C for 3 days. **d** Tenfold serial dilutions of exponentially growing cells with the indicated mutations were spotted onto YPD plates (+CPT, HU, or MMS) and incubated at 30 °C for 3 days. Different concentrations of CPT, HU and MMS were used depending on strain sensitivity. **e** Tenfold serial dilutions of exponentially growing cells with the indicated mutations were spotted onto YPD+HU 30 mM plates and incubated at 30 °C for 3 days. Source data are provided as a Source Data file.

was not until recently that it started to be studied in the context of replication fork stalling. The Set1C histone methyltransferase in yeast is known to be involved in DSB repair[32], prevention of TRC[33] and recovery of HU-arrested forks[35]. Growing evidence supports the notion that the H3K4 methyltransferase family acts differently according to replication stress type[36,37].

Here, we used the timely-inducible and site-specific Tus/*Ter* replication fork barrier downstream of ARS305 to study the role of Set1C. By combining 2D-gels and Cdc45 ChIP analysis, we found that the Tus/*Ter* barrier causes efficient and transient replication fork stalling, consistent with previously published data[41–43]. Interestingly, we reveal that Tus is dislodged to some extent from the *Ter* sites during fork stalling. This observation supports the "Tus-*Ter* lock" model caused by the helicase unwinding of the *TER* sequence.

Surprisingly, we found that Spp1 (but not Set1) is recruited to the stalled replication fork. Spp1 binding occurs independently of Set1C whose recruitment to chromatin depends on RNA PolII and Swd2[46–49]. Indeed, active transcription in S-phase was not detected upstream of the stalled fork, and Swd3 was not detected at the barrier. We further report that Spp1 recruitment upstream of the stalled fork depends on its PHD finger domain. Spp1 binding to H3K4me3 via its PHD finger domain was previously reported during meiosis[44,45,50,51]. Analysis of Spp1 recruitment in the *swd3Δ* mutant could not be used to demonstrate that Spp1 is recruited independently of Set1C, as *SWD3* deletion completely abolishes H3K4 methylation, and this will affect per se the interaction between the Spp1 PHD domain and the nucleosomes that will be unmethylated at H3K4. Although, we rationalized that Spp1 could be recruited to stalled forks via its PHD domain independently of its interaction with Set1C, its recruitment to chromatin at the barrier is indeed H3K4me3-dependent and thus Set1 dependent. Once Spp1 is on the chromatin, it acts independently of Set1C.

Interestingly, during chromatin replication, the recycled parental histones are marked by H3K4me3[16,18–20]. We found that H3K4me3 is distributed upstream of the barrier, and most importantly, Spp1 occupancy overlaps with H3K4me3 parental histone mark during fork stalling. We propose that parental histones marked by H3K4me3 serve as a docking site for Spp1 recruitment to Tus/*Ter* barrier. Of note, we observe parental mark dilution as a consequence of DNA replication as previously described[17,20]. Recently, it has been reported that Spp1 is recruited at weakly transcribed regions via its PHD domain to allow restoration of the H3K4me3 mark in cells having asymmetric distribution of parental histones[52]. These findings support our observations that Spp1 is recruited to chromatin at regions with minimal H3K4me3 independently of Set1C. In our experiments, neither Set1 nor an increase in H3K4me3 was detected at the time of replication fork stalling. Our data suggest that Spp1 reads H3K4me3 behind the stalled replication fork independently of its association with Set1C.

We further report that upon *SPP1* deletion, the replication fork stalls at earlier time points with a stronger stalled fork signal at 30 min at the Tus/*Ter* barrier. Intriguingly while the signal was stronger at 30 min, it was almost undetectable at 40 min. In contrast, the stalled fork signal was weaker in WT but persisted until 40 min. Molecular combing of DNA fibers from spp1Δ cells grown in S-Raffinose suggests that replication forks travel faster in the absence of Spp1, at least, in the conditions used for our experiments. These observations, combined

with the Cdc45 ChIP experiments, indicate that the absence of Spp1 results in faster and stronger replication fork pausing at Tus/*Ter* barrier. We also observed that the absence of Spp1 led to a more synchronized progression that could explain the strong signal of the stalled fork. Nonetheless, we think that the strong stalling signal may reflect the requirement for longer fork processing in the absence of Spp1.

In WT cells, the Tus/*Ter* barrier system causes transient replication fork stalling without inducing DNA breaks[41,42]. It is demonstrated that the interplay between fork remodelers (Mph1FANCM/Rad5HTLF), resection machinery (Exo1/Dna2), homologous recombination machinery (Rad51/Rad52/Rad59) and helicases (Sgs1, Srs2) allows error free fork recovery[43]. Similarly to WT, we found that in spp1 mutants, there is Exo1-dependent resection at the stalled fork. However, we found excessive RPA binding within *Ter* repeat only in spp1 mutants, thereby suggesting that Spp1 restricts the availability of DNA to nucleases.

Consistent with all the results that we have obtained, we observed a significant increase in mutagenesis in spp1Δ cells. It was shown that Tus/*Ter*-dependent fork stalling causes mutagenesis due to fork slippage or misalignments during strand invasion[42]. This mechanism may still operate in spp1Δ cells. However, the fact that microdeletions (<3 bp) were increased in both *radd52Δ and spp1Δ* cells suggests that a mechanism other than HR is responsible for the formation of the microdeletions. We propose that NHEJ could operate in the absence of either *RAD52* or *SPP1*. The fact that the mutation rates observed in each of the single mutants are additive in the double mutant suggests that two parallel mechanisms are at work in the double mutant. Interestingly, it was shown in *S. pombe* that Rad52 also protects arrested forks by limiting the activity of Exo1 at the RTS-RFB barrier[66].

During fork perturbation, fork reversion could lead to different outcomes depending on the downstream factors[9,11,14,57]. For instance, fork reversal generates a DNA structure that is protective against nucleolytic degradation; however, the absence of protective factors creates an entry point for nucleases. Along this line, SETD1A, the close homolog of Set1C, was shown to protect the nascent DNA from excessive resection[37]. Surprisingly, we found that drivers of fork reversal in yeast, Rad5 and Mph1, are essential during replication stress in the absence of Spp1, suggesting that reversion of the stalled fork becomes an important event in the absence of Spp1. Indeed, we found that chromatin organization is altered in cells lacking Spp1 upstream of the replication fork stalling, suggesting an increase in chromatin remodeling. We propose that Spp1 binding to the nascent chromatin upstream of the Tus/*Ter* barrier creates a protective chromatin environment that subsequently limits fork remodeling. In contrast, the absence of Spp1 results in a fork environment that is more prone to remodeling at the level of both the fork and chromatin. Nonetheless, increased remodeling of the stalled fork at the barrier leads to DNA exposure to nucleases. Thus, this could explain why in the absence of Spp1, the *Ter* array is covered with RPA. Simply, Spp1 binding to nascent chromatin could subsequently prevent or delay the recruitment of chromatin remodelers or histone acetyltransferases, therefore, promoting a balance between chromatin organization, fork remodeling and fork degradation. It is not clear the reason behind the difference between *SPP1* and *SET1* deletion on nascent DNA resection[35]. One way to explain the difference is that the balance between methylation

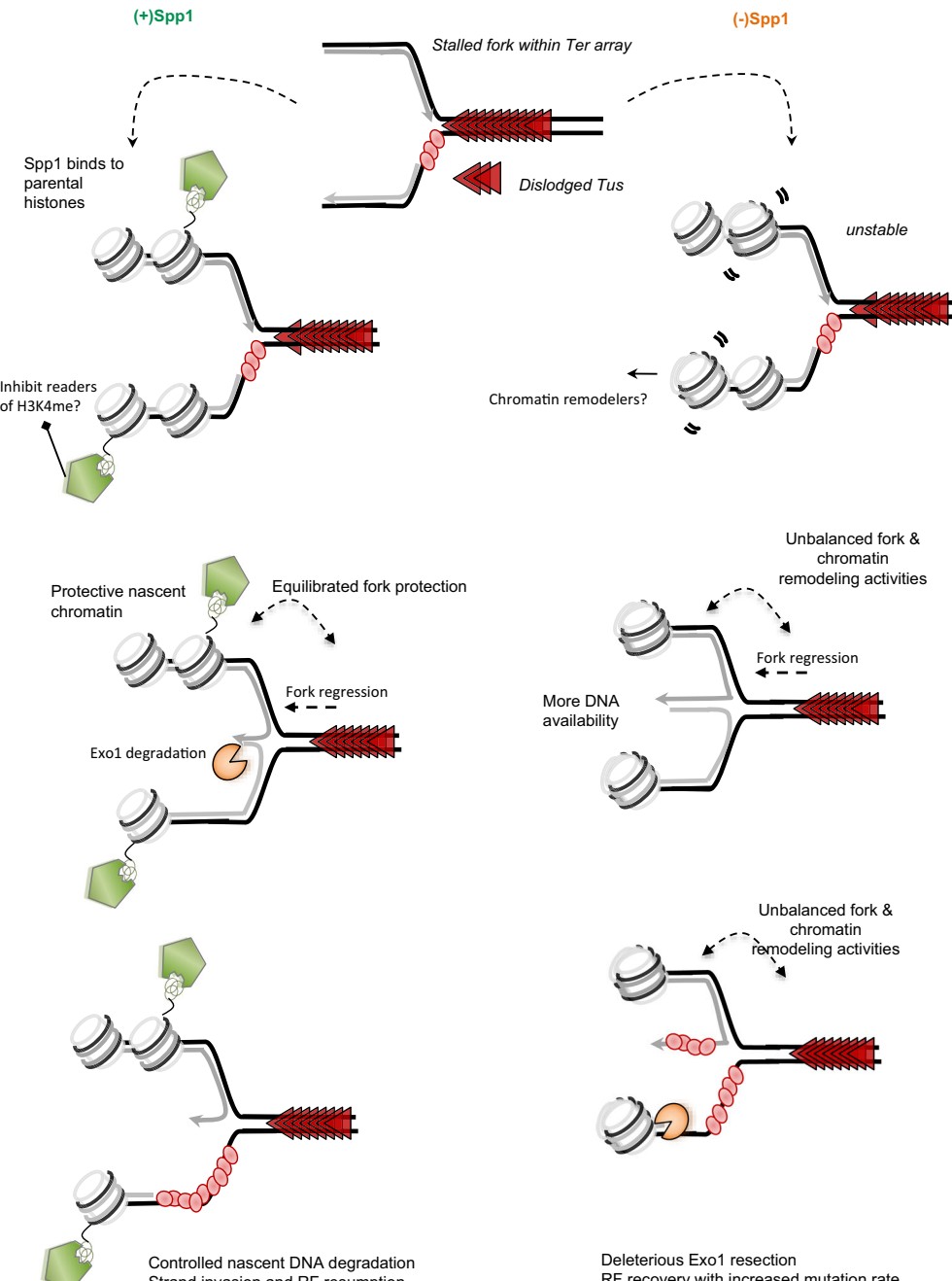

**Fig. 9 | Model of Spp1-mediated nascent DNA protection against excessive nucleolytic degradation at the Tus/*Ter* barrier.** As the replication fork stalls, the H3K4me3 parental histones recruit Spp1 via its PHD finger domain to protect chromatin. Spp1 binding to nascent chromatin could directly restrict fork remodeling activity or/and delay the recruitment of H3K4me readers such as chromatin remodelers and modifiers. The equilibrium between chromatin and fork remodeling ensures fork protection as well as the regulation of fork resection by Exo1 to allow proper fork resumption. In the absence of Spp1 (right), replication fork stalling causes unbalanced fork and chromatin remodeling, increasing nascent DNA accessibility. This creates more DNA availability and an entry for Exo1-dependent resection. RPA becomes essential to protect the exposed ssDNA and allow repair. In both cases, the replication fork resumption will take place after strand invasion and annealing within *Ter* array.

patterns can ultimately affect the choice between either increasing or restricting DNA accessibility in combination with which downstream affecter is in play. Even though the exact mechanism of Spp1 protecting nascent DNA at the Tus/*Ter* barrier is still to be fully understood, we propose that, as the replication fork stalls at the replication fork barrier, Spp1 is recruited via its PHD domain to nascent chromatin and creates a balanced chromatin environment in terms of fork protection and processing that subsequently prevents deleterious nucleolytic degradation (Fig. 9). In conclusion, our findings demonstrate a new function and recruitment of Spp1 at a single protein/DNA

barrier and its importance in protecting the stalled fork from toxic degradation. Our data further reinforce the view that the chromatin environment is an important regulator of the replication stress responses.

Although all the mechanisms we have described regarding the role of Spp1 at the stalled replication fork apply to the Tus/*Ter* barrier, we found that combining Spp1 loss with RPA mutants is lethal or strongly deleterious for the cells in the presence of genotoxic agents. These data suggest that Spp1 may have a genome-wide role in protecting against excessive ssDNA formed during replication stress.

## Methods

### Strains and plasmids

All yeast strains used in this study are isogenic derivatives of *W303* listed in Supplementary Table S1. The 21 bp *TerB* repeats were amplified by PCR from plasmids pNBL63 (restrictive orientation) and integrated 3 kb downstream ARS305, as described in ref. 41. The plasmids p415-$P_{GAL1}$-HA-*Tus* or p415-$P_{GAL1}$ (empty vector) are transformed into the cells to induce the barrier system. For cell sensitivity to genotoxic drugs, HU and CPT and MMS were added to the media on plates.

### Cell growth and synchronization

Cells were grown at 25 °C in S-Raffinose unless otherwise indicated. Exponentially growing cells were synchronized in G1 using 8 µg/ml α-factor for 3 h. Tus expression was induced by adding 2% Galactose (final w/v) for the final 2.5 h of the G1-arrest. Release of cells from G1-arrest was achieved by centrifugation, washing and resuspension of cells in pre-warmed fresh medium. The same condition was used for EdU-FACS kinetics with few modifications. EdU pulse was added for 15 µM final concentration after release from G1-arrest for 20 min and then chased with 10X thymidine.

### Chromatin immunoprecipitation (ChIP)

Chromatin samples were prepared as previously described using the following steps[67]. Cells were crosslinked with 1% formaldehyde for 15 min, followed by 5 min quenching with 1.5 M glycine. Cells were lysed by vortexing with glass beads (30 × 30 s, with cooling between cycles, in lysis buffer (50 mM HEPES-KOH [pH 7.5], 140 mM NaCl, 1 mM EDTA, 1% Triton X-100, 0.1% sodium deoxycholate, supplemented with protease inhibitors). Cell debris was removed by microcentrifugation, and the chromatin sheared to ~200 bp using a Bioruptor Pico sonicator. Insoluble material was removed by microcentrifugation for 10 min at 14,000 rpm at 4 °C.

For immunoprecipitation, 500 µg of chromatin was incubated with the following antibodies: 3.5 µl Anti-PK (anti-V5 tag) Life Technologies Cat#R960-25, 2 µl Anti-HA Santra Cruz Cat#SC-7392, 2.5 µl Anti-Myc (9E10) Santa Cruz Cat#sc-40, 2.5 µl Anti-RPA Agrisera Cat#AS07214, 1 µl Anti-H3 Abcam Cat#Ab1791, 1 µl Anti-H3K4me3 EpiGenetek Cat# A-4033-100, 1 µl Anti- RNA pol II CTD phospho Ser5 Active Motif Cat#61086. Then the lysate was incubated with 25 µl of Protein G-Sepharose beads at 4 °C overnight in FA lysis buffer. Precipitates were washed once with lysis buffer and twice with Lysis buffer having 500 mM NaCl. The beads were then washed twice with Wash Buffer (10 mM Tris-HCl [pH 8.0], 0.25 M LiCl, 1 mM EDTA, 0.5% NP-40, 0.5% Na-Deoxycholate), and once with TE (10 mM Tris-HCl [pH 8.0], 1 mM EDTA) buffer. Precipitated materials were eluted with buffer containing 50 mM Tris-HCl [pH 7.5], 10 mM EDTA and 1% SDS by incubating at 65 °C for 10 min. Subsequent decrosslinking was performed at 65 °C overnight. DNA was purified using MSB® Spin PCRapace Kit. Oligonucleotides used for qPCR reactions are listed in Supplementary Table S2.

### Flow cytometry analysis

One milliliter of cells was harvested by centrifugation and then fixed in 70% ethanol overnight. Cells were washed and resuspended in 1 ml of 50 mM Tris-HCL (pH 7.0). Cells were briefly sonicated and then treated with 0.25 mg/ml RNase A for 1 h at 50 °C. Proteinase K was then added to a final concentration of 1 mg/ml, and cells were incubated for a further 1 h at 50 °C. Samples were then diluted in 50 mM Tris-HCL with 0.5 µM Sytox green and incubated at room temperature for a minimum of 30 min. Samples were analyzed using a Becton Dickinson BD Accuri™ C6 Plus machine, using BD CSampler™ Plus Software. FACS for EdU-labeling experiments were performed as previously described[53].

### 2D gel analysis of DNA structures

For this, 600 ml aliquots of cells were killed by the addition of 0.1% (final w/v) sodium azide at defined time points and harvested by centrifugation. In vivo psoralen crosslinking and 2D gel analysis have been described[68,69]. DNA was purified using Qiagen Genomic 100G Tip extraction kit.

For each 2D gel image, 20 µg of DNA was digested overnight with the indicated restriction enzymes (Xho*I*). The DNA was ethanol-precipitated and resuspended in 20 µl of Tris-EDTA buffer. Samples were run on 0.4% low EEO agarose (US Biological, USA) first-dimensional gels at 50 V for -16 h and then stained with 0.3 µg/ml ethidium bromide. Gel strips were cut from first-dimensional gels and run on 0.90% agarose second-dimensional gels at 180 V (in Tris-borate-EDTA buffer containing 0.3 µg/ml ethidium bromide) for ~8 h. DNA was transferred to Genescreen Hybridization Transfer Membranes (Perkin Elmer, USA) by southern blotting, and the DNA was immobilized by ultraviolet crosslinking. DNA replication intermediates present at ChrIII were detected using unique chromosome-specific $^{32}$P dCTP (6000 Ci /mol; Perkin Elmer)-radiolabelled probes that were synthesized using the Rediprime II kit (GE Healthcare, Denmark). Stripping of membranes for subsequent reprobing was achieved by washing the membranes with a boiling solution of 0.1% SDS. Quantification of signals was performed using Image Quant analysis software (Molecular Dynamics, Sunnyvale, CA).

### Analysis of mutation rates and types

Cells were grown to saturation in an S-Raffinose containing 2% galactose medium for analysis of mutation rate and types, then serial dilution of the culture was done, and each dilution was plated onto nonselective plates. Plates were incubated at 30 °C for 2–3 days and then replica plated onto plates containing 5-FOA. Mutation rates were measured by fluctuation analysis[70,71]. Individual colonies were confirmed as 5-FOA resistant, and the *URA3* locus was sequenced using primer covering the *URA3* gene. Statistical analysis of differences in mutation rates was performed using a one-sided Mann–Whitney U test.

### Molecular combing

Cells are grown in S-Raffinose (+Gal) and pulsed with 25 µM EdU for 20 min, 10 ml of each sample is added to 40 ml ice-cold TE$_{50}$. Cells are washed twice with cold TE. Pellet is then resuspended in 1 ml NZ buffer (1.2 M Sorbitol, 50 mM citrate phosphate buffer pH5.5, 50 mM EDTA) and transferred to 2 ml Eppendorf tubes, pelleted at 4 °C. Pellets are then resuspended in 250 µl Zymolyze buffer (1 ml NZ buffer+ Zymolyze+ 10 mM DTT) and 150 µl low melting point agarose gel (LMP). Using a P1000 pipette, the mix is poured into a plug apparatus. The plugs are put in a warm chamber at 37 °C for 1 h, then transferred to room temperature for 10 min and then at 4 °C for 20 min. The solidified plugs are then transferred into PK buffer (125 mM EDTA pH 9.5, 1% Sarkosyl, 1 mg/ml Proteinase K) in a 15-ml round-bottomed tube and put at 37 °C overnight. A fresh PK buffer is added for two consecutive days. After PK washes, the plugs are washed twice in 1X TE for 2 h. One plug is transferred to a new tube and washed twice with 100 mM NaCl for 30 min. The plug is then incubated for 5 min with 50 mM MES and 100 mM NaCl, then transferred to a fresh buffer and incubated for 45 min at 68 °C. When the plug is completely melted, cool down at 42 °C and add 3U/Plug of beta-agarase and leave overnight. The second day 1U/plus of beta agarose is added for 2 h; afterward temperature is increased to 65 °C for 10 min and the DNA samples are stored in the dark until combing. DNA was combed on Genomic Visions Cover Slips and then baked at 65 °C for 2 h and let in the dark overnight. Slides are incubated with PBS-Triton 0.1% BSA 1% for 35 min, then washed with PBS-triton before doing the Click reaction to visualize the EdU-labeled DNA. The EdU detection was done according to the protocol from

Thermo Fisher (Invitrogen) kit for Click reaction. The reaction was done twice, and then antibody against autoanti-ssDNA (DSHB, AB_10805144) with a dilution of 1/50 in PBS-Triton-BSA buffer and incubated at 37 °C for 1 h. The slides are then washed trice on PBS-Triton and incubated with 1/50 Alexa Fluor 647 (Thermo Fisher, Cat#A-21241) and incubated for another 1 h at 37 °C. The slides are then washed trice and let to be dried completely before adding 8 µl Prolong gold and covered by a protective cover slip, left overnight in the dark. The slides are analyzed using Metamorph imaging software.

**MNase digestion and Nascent chromatin accessibility assay**
For MNase accessibility to nascent chromatin, cell growth and synchronization were done as previously described, except cells were released in the presence of 25 µM EdU for a pulse of 20 min, and samples were collected at 20, 30 and 40 min after release from alpha factor. The budding index was also considered before collection to ensure that the time points were comparable between different backgrounds. Cells were crosslinked and frozen until further experimentation. Samples were split to perform MNase digestion, undigested DNA and H3 ChIP. Here MNase amount was adjusted to each spheroplast sample size to obtain 70–80% mononucleosomes before purifying the 150 bp mononucleosomes from 2% agarose gel. MNase digestions were performed as previously described[72].

**Click reaction and streptavidin affinity capture**
Click reaction and streptavidin capture were done as described in refs. 55,56 with few modifications. Briefly, MNase-digested DNA or sonicated (to 150–200 bp) were incubated in a click chemistry reaction buffer. The click reaction proceeded for 1–2 h at room temperature with gentle shaking. DNA was recovered by ethanol precipitation and resuspended in 1X TE. Biotin-conjugated EdU-labeled DNA was incubated with streptavidin-coated beads (previously blocked with salmon sperm DNA and washed twice with cold lysis buffer). The incubation was performed overnight at 4 °C. Bead-bound DNA was washed three times with wash buffer. DNA was eluted using 1% SDS-TE and incubated at 95 °C for 15 min. The recovered DNA was analyzed by qPCR.

**Reporting summary**
Further information on research design is available in the Nature Portfolio Reporting Summary linked to this article.

## Data availability
The authors state that all data necessary for confirming the conclusions presented in the manuscript are represented fully within the manuscript or in the manuscript tracking system as Source data Excel tables. Source data are provided with this paper.

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

## Acknowledgements

We thank Hocine Mankouri and Ian Hickson for providing the Tus/*Ter* barrier system. We thank Giulia Mazzucco (IFOM-ETS) for her help with 2D gels and Armelle Lengronne for the general discussion about the work, providing the strain for incorporating thymidine analog and

plasmid for V5 tagging and her help with the DNA combing technique. We thank Marie-Noelle Simon, Marion Dubarry and Christelle Cayrou for their discussions about the work. V.G. laboratory is supported by the "Ligue Nationale Contre le Cancer" (LNCC) (Equipe labellisée). N.G. is supported by "Bourse du Ministère de l'Enseignement Supérieur, et de la Recherche (MESR)" and by « Fondation pour la Recherche Médicale (FRM)».

## Author contributions

N.G. performed most of the experiments presented in this study. Y.C. performed Fig. 8 and P.L. contributed to several experiments. N.G., Y.C. and V.G. designed experiments and analyzed the data. M.G. and Y.D. designed and performed the 2D-gel analyses. N.G. and V.G. wrote the paper.

## Competing interests

The authors declare no competing interests.
