## [Peer Review File · Nature Communications]

The COMPASS subunit Spp1 protects nascent DNA at the Tus/*Ter* replication fork barrier by limiting DNA availability to nucleasesREVIEWER COMMENTS

Reviewer #1 (Remarks to the Author):

Ghaddar et al., report the finding that The SetC Spp1 subunit is recruited to the single, site-specific replication barrier—the Tus/Ter complex in a PHD domain dependent manner, independently of Set1c and that Spp1 seems to restrict Exo1-dependent ssDNA formation at Tus/Ter barrier. These indicated that Spp1 indeed plays a role in such a site-specific replication barrier. However, the author missed out detailed experiments to mechanistically define Spp1's role at this replication fork barrier. Besides, the manuscript is missing critical analysis that are necessary to evaluate the conclusions of the authors: Spp1 protects nascent DNA at stalled replication fork barrier by limiting DNA availability to nucleases. More genome-wide data are required to connect the finding obtained from the studies at the artificial replication fork barrier to the conclusion the author eventually drew at the genome-wide scale.

- Page 6 Line 147 & 156 & 164 & 179 & 192: The general conclusion, Spp1 recruitment to chromatin as RF stalls, cannot be drawn from these studies. The data can only suggest Spp1 recruitment to the single, site-specific replication barrier (the Tus/Ter complex)
- Page 8 Line 203 (whole paragraph): (1) The author should provide additional experiments to show the localization of Spp1 observed at the single, site-specific replication barrier can really mirror what happens at stalled replication forks in the genome-wide scale. For example, the cut and run assay to map Spp1's localization upon replication stress. (2) no clear explanation for the rationale of faster RF progression caused by spp1Δ.
- Page 8 Line 213: The term "G1-S phase transition" used should be more careful. The data can only be interpreted as faster S phase progression and the conclusion "an enhanced G1-S phase transition" should not be drawn unless the author performs other specific assay to prove it.
- Page 9 Line 247: The author concluded that Spp1 restricts the Exo1-dependent degradation. However, an Exo1-generated ssDNA gap behind the RF stalling at Tus/Ter Barriers has been shown to be a prerequisite step for subsequent homologous recombination (ref. 44). Seemingly, these two conclusions are contradictory. Although the author claimed that Spp1 restricts ssDNA formation by promoting protective nascent chromatin (figure 8), I am not quite convinced by the implication drawn by the experiment. It seems like Spp1 is involved in a timing mechanism that limits the extent of DNA end resection for the proper downstream response under this specific stress. The author did not mechanistically define Spp1's role in this aspect. To clarify the role and also emphasize the physiological importance of Spp1, the author should provide more insights into the consequence of the Tus/Ter induced stalled replication fork when Spp1 is deleted using 2D gel or/and DNA combing analysis.
- Page 9 Line 258: Please provide the reference for the statement "The mutation rate increase is one of the hallmarks of increased ssDNA formation".
- Page 9 Line 261: In Fig 6b, the author tried to use the conclusion obtained from the studies of a single, site-specific replication barrier as the rationale for the genome-wide studies. Before doing so, I would expect the experiments to show RPA is colocalized with Spp1 globally upon replication stress and an increase of global RPA-ssDNA in spp1 mutants is "stress-dependent".
- Page 10 Line 271: Two questions for Figure 6c: (1) How about challenging the mutants with other replication stressors (HU and MMS)? (2) Can the deletion of EXO1 rescue the rtt105Δ spp1Δ?
- Page 10 Line 279: The word "essential" used is not proper as the pCLB2-RFA1 spp1Δ cells did not die upon CPT. Instead, the word "critical" should be used.
- Page 10 Line 288: the same problem as Figure 6b—the author missed out showing the increase of Rad52 foci in spp1Δ cells was indeed caused by replication stress.
- Page 11 Line 313: In Fig 8b, what was the rationale for the author to choose "CPT, 30 min" as a replicative stress condition. If it is a right condition, should one expect a similar histone H3 ChIP profile in this situation as that in the Figure 7b?
- Page 11 Figure 8: The whole experiment setup should be in a genome-wide scale if the author expected to draw the final conclusion—Spp1 binding to chromatin is important for nascent DNA protection during RF stalling. The MNase-seq experiment should be pursued.

Minor points:

- Some figure labels should be checked:
 - Page 6 Line 142

- Page 6 Line 144
- Page 6 Line 156
- Page 7 Line 170:
- Page 10 Line 288
- Page 5 Line 107: RFB: replication fork barrier
- Page 5 Line 128: RF: replication forks
- Page 10 Line 280: RS: replication stress
- Page 7 Line 186: The sentence should be checked
- Page 8 Line 214: SPP1 mutants \diamond spp1 mutants

Reviewer #2 (Remarks to the Author):

This manuscript from the Geli laboratory reports that the Spp1 protein of *Saccharomyces* is involved in preventing DNA at stalled DNA replication forks from being processed by Exonuclease 1. Further, they show that Spp1 is recruited to chromatin through an interaction with histones carrying the H3K4me3 mark. There is considerable interest in understanding the steps that occur downstream of replication fork stalling and this study utilizes a neat system that creates a locus-specific block in the yeast genome. The results would be of interest to labs working on DNA replication, repair, and chromatin structure, but in its present form it has some weaknesses that I would recommend be addressed prior to publication.

Major Points

1. The standard of the English would benefit considerably from careful editing and proofreading.
2. There are two issues with the 2D gels presented in Figure 1. First, the modest quality of the image makes it impossible to discern where the Y-arc and other characteristic structures lie. The X-shaped molecule is also poorly visible. To make the data convincing, the authors should run gels with samples taken at different time points after release from alpha-factor, including much earlier time points than are shown, and couple these with flow cytometry traces at each time. Second, it is claimed that the X-shaped molecules are recombination intermediates. This cannot be claimed without analysis of a rad51 or rad52 mutant.
3. The timing of appearance of the replication fork at the barrier seems unusually late. As the barrier is adjacent to ARS305, which fires immediately after S-phase entry, it is hard to imagine that it takes 40 mins after alpha-factor release for replisome components to be detected at the barrier. What is the possible explanation for this? Previous work on the Tus-Ter barrier has shown that the replisome is maximally arrested by about 30 mins, after which the transient arrest starts to be overcome. Is it possible that the authors have missed the peak of binding because of using time points that are too late? Please also see a related comment in point 8 below.
4. To make claims about the speed of the fork movement being faster in spp1 mutants, it is necessary to do the analysis more rigorously by doing 2d gels probed with different probes from along the genomic region in a time course experiment. One possible explanation for the effects seen is not that the replisome speeds up, but instead that the spp1 mutant cells enter S-phase more rapidly due their size. Any mutant cell that is unusually large tends to fulfill the G1 size checkpoint very quickly and race into S-phase after release from alpha-factor.
5. Previous studies indicated that recombination mechanisms generated many of the Tus associated mutations. This dependence should be tested here too on the mutations in spp1 cells.
6. There is a lack of consistency in how the cells are treated. In some experiments camptothecin is added, but then on page 12, HU is added. There was no explanation for the use of different compounds in these experiments.
7. Has it been tested whether induction of the barrier alone is sufficient to kill an mph1 rad5 spp1 triple mutant, rather than using HU?
8. I found the EdU data in Fig 4c confusing. In the WT cells, there was only minor amounts of EdU incorporated before the 45 min time-point and yet the flow cytometry traces in Fig 3 indicate that the WT cells are in mid S-phase and in many cases even in G2 by at an earlier time-point (40 mins) after release.

Minor Points

1. The authors should standardize on calling the complex COMPASS or Set1C. Both names are used throughout.
2. There are several overstatements and conclusions that, while perhaps correct, are only one possible explanation. It would be preferable to see these toned down a bit. For example, on page 9 it is stated that 'Spp1 restricts nucleolytic degradation of nascent DNA'. I don't think it was shown that it is nascent DNA that is being degraded. Also, on page 9 'The mutation rate increase is one of the hallmarks of excessive ssDNA formation this, these results confirms that Spp1 leads to excessive ssDNA formation as RF stalls'. This sentence is ungrammatical and it is also an overstatement.

POINT BY POINT RESPONSE TO REVIEWER COMMENTS

We thank the referees for their constructive comments. Before responding point by point to the referees' comments, we would like to make a general comment on the reorganization of the revised manuscript.

The main purpose of using Tus/*Ter* barrier is to look at site-specific replication fork stalling. While, in principle, it should reflect a stalled fork, the nature of stalled fork may differ according to the nature of the replication stress. Tight DNA-protein complexes certainly represent one of the most physiological replicative stresses that cells must face.

We would like to point out that dissecting the genome-wide role of Spp1 at natural protein/DNA barriers is something very difficult to achieve due to the fact that Spp1 is part of the Set1-Complex (Set1C or COMPASS) whose binding to chromatin is mediated in part by its interaction with RNA-POLII (See our MS, Bae et al. 2020, Nature Communications). We have performed ChIP-Seq of Spp1 and Spp1^{ΔPHD}. The genome-wide occupancy of Spp1 to chromatin mirrors the one the Set1-C and H3K4me3. When we compared Set1 and Spp1 peaks, it was very difficult to detect specific Spp1 peaks that stood out from the general binding profile of Set1 (associated with transcription). Therefore, the role of Spp1 at natural protein/DNA barrier using genome-wide approaches such as ChIP-seq or cut and run assay is made very difficult by the tight association of Spp1 to the Set1C. The Chip-Seq profile of Spp1^{ΔPHD} is similar to the one of Spp1 (and Set1), except that the peak intensities were reduced by approximately 30%.

We therefore have reorganized the revised manuscript to emphasize that our conclusions apply essentially to the single, site-specific replication barrier (the Tus/*Ter* complex) as demanded by referee 1.

- The first 3 and the 5th sections remain unchanged.
- We focus in the 4th section in the dynamics of the Tus/*Ter*-dependent replication fork stalling in the *spp1* mutants. We provide new 2D gels showing an earlier and stronger replication fork stalling at the Tus/*Ter* barrier in *spp1Δ* cells. In addition, in this section we have monitored genome-wide replication fork progression by performing molecular combing of DNA fibers from asynchronous WT and *spp1Δ* cells. The results show that the replisome moves faster in *spp1Δ* cells grown under the conditions of our experiments (S-Raff + GAL).
- In the 6th section we have extended the study of mutagenesis upstream of the Tus/*Ter* replication barrier and studied the genetic dependence of the mutagenic process induced by the absence of Spp1.

- In the 7th section of the manuscript, we now investigate the nascent chromatin organization upstream of the Tus/Ter barrier (and not at the vicinity of ARS) by analyzing the nascent chromatin protection against MNase digestion. We observed a strong decrease of chromatin protection in the absence of Spp1 upstream of the Tus/Ter barrier.
- Finally, we have gathered the data showing that deletion of *SPPI* sensitizes cells to RPA dysfunction in a final section.

By doing so, we hope to clarify that the data apply to the Tus/Ter replication fork barrier. We provide here below our point-by-point response to all the concerns raised by referees.

Reviewer #1 (Remarks to the Author):

Ghaddar et al., report the finding that The Set1C Spp1 subunit is recruited to the single, site-specific replication barrier—the Tus/Ter complex in a PHD domain dependent manner, independently of Set1c and that Spp1 seems to restrict Exo1-dependent ssDNA formation at Tus/Ter barrier. These indicated that Spp1 indeed plays a role in such a site-specific replication barrier. However, the author missed out detailed experiments to mechanistically define Spp1's role at this replication fork barrier. Besides, the manuscript is missing critical analysis that are necessary to evaluate the conclusions of the authors: Spp1 protects nascent DNA at stalled replication fork barrier by limiting DNA availability to nucleases. More genome-wide data are required to connect the finding obtained from the studies at the artificial replication fork barrier to the conclusion the author eventually drew at the genome-wide scale.

See our general comment.

- Page 6 Line 147 & 156 & 164 & 179 & 192: The general conclusion, Spp1 recruitment to chromatin as RF stalls, cannot be drawn from these studies. The data can only suggest Spp1 recruitment to the single, site-specific replication barrier (the Tus/Ter complex)

We have reorganized the MS and rephrased the sentences to highlight that the observations we made are specific to the Tus/Ter barrier. The title and abstract were changed accordingly.

- Page 8 Line 203 (whole paragraph): (1) The author should provide additional experiments to show the localization of Spp1 observed at the single, site-specific replication barrier can really mirror what happens at stalled replication forks in the genome-wide scale. For example, the cut and run assay to map Spp1's localization upon replication stress.

See our general comment.

(2) no clear explanation for the rationale of faster RF progression caused by *spp1Δ*.

The faster RF progression caused by *spp1Δ* appears to be specific to the raffinose carbon source used in the settings of our experiments. We provide in the revised version of the manuscript DNA combing results showing that asynchronous *spp1Δ* cells display longer tracks of EdU in S-Raff (+GAL) compared to WT (New Figure S4) suggesting that the replisome indeed moves faster.

We observed that raffinose damages DNA, particularly in the *spp1Δ* mutant (new Figure 6a). This is likely due to the production of high levels of ROS produced by the mitochondria. It is possible that the higher level of DNA damage in the *spp1Δ* mutant grown in S-Raff (+GAL) increases the levels of dNTPs (through the induction of RNR3). As a result, this would increase the fork rate in the *spp1Δ* mutant. This explanation is provided in the discussion of the MS.

- Page 8 Line 213: The term “G1-S phase transition” used should be more careful. The data can only be interpreted as faster S phase progression and the conclusion “an enhanced G1-S phase transition” should not be drawn unless the author performs another specific assay to prove it.

Using EdU FACS we can clearly see that cells started to incorporate EdU earlier reflecting an earlier entry to S phase. We have shown these data to try to explain the differences in RPA and Cdc45 detection at Tus/Ter barrier. While there can be multiple reasons behind this, we think it is due to the experimental conditions we’ve been working on (growing cells in S-Raffinose). We have rephrased the sentences to avoid confusion.

- Page 9 Line 247: The author concluded that Spp1 restricts the Exo1-dependent degradation. However, an Exo1-generated ssDNA gap behind the RF stalling at Tus/Ter Barriers has been shown to be a prerequisite step for subsequent homologous recombination (ref. 44). Seemingly, these two conclusions are contradictory. Although the author claimed that Spp1 restricts ssDNA formation by promoting protective nascent chromatin (figure 8), I am not quite convinced by the implication drawn by the experiment. It seems like Spp1 is involved in a timing mechanism that limits the extent of DNA end resection for the proper downstream response under this specific stress. The author did not mechanistically define Spp1’s role in this aspect. To clarify the role and also emphasize the physiological importance of Spp1, the author should provide more insights into the consequence of the Tus/Ter induced stalled replication fork when Spp1 is deleted using 2D gel or/and DNA combing analysis.

Our data show that deleting full length *SPP1* or its PHD domain increases the Exo1-generated ssDNA gap behind the RF stalled at the Tus/Ter barrier. Generation of ssDNA is observed to a lower extent in WT cells. We do not know the exact mechanism by which Spp1 limits the Exo1-dependent generation of ss-DNA. As asked by referee 1, we have performed 2D-gel

analysis in *spp1Δ* cells (new Fig. 4b) showing that the RF arrest induced by the barrier occurs earlier and is exacerbated in *spp1Δ* cells.

- Page 9 Line 258: Please provide the reference for the statement “The mutation rate increase is one of the hallmarks of increased ssDNA formation”.

We apologise, we have added the references.

- Page 9 Line 261: In Fig 6b, the author tried to use the conclusion obtained from the studies of a single, site-specific replication barrier as the rationale for the genome-wide studies. Before doing so, I would expect the experiments to show RPA is colocalized with Spp1 globally upon replication stress and an increase of global RPA-ssDNA in *spp1* mutants is “stress-dependent”.

Following the recommendation of Referee 1, we first tried to show co-localisation of RPA and Spp1 foci by fluorescence microscopy in camptothecin (CPT) treated cells. However, the Spp1-GFP signal was rather diffused in the nucleus and hard to visualise, either in the absence or presence of CPT while clear RPA foci were observed in CPT treated cells. As mentioned in our general comment, this is most probably due to the fact that a large fraction of Spp1 is associated with the Set1C on the chromatin.

Second, we have integrated new data showing that there is an increase of Rfa1-CFP and Rad52-YFP foci in *spp1Δ* cells treated with CPT compared to WT-treated cells (New Figure 6a). Of note the foci are much brighter in the absence of Spp1. These results indicate that the increase of global Rfa1-CFP and Rad52-YFP in *spp1Δ* cells is indeed “stress-dependent”.

- Page 10 Line 271: Two questions for Figure 6c: (1) How about challenging the mutants with other replication stressors (HU and MMS)? (2) Can the deletion of EXO1 rescue the *rtt105Δ spp1Δ*?

(1) We exposed *spp1Δ*, *rtt105Δ*, and the double *spp1Δ rtt105Δ* to HU and MMS. Deleting *SPP1* in *rtt105Δ* cells has mainly an effect in CPT treated cells. This is why we show only the CPT-treated cells in Figure 6c (now Figure 8b).

(2) We have performed the experiment (shown in Figure S7). Deleting *EXO1* did not rescue the *rtt105Δ spp1Δ* sensitivity to CPT. We show that Exo1 is majorly responsible for ssDNA generation at the Tus/Ter barrier in the absence of Spp1, consistent with previous results showing that Exo1 generates ss-DNA at the barrier in WT cells. However, this does not preclude the role of other nucleases in processing genome-wide damage generated in the *spp1Δ rtt105Δ* double mutant.

- Page 10 Line 279: The word “essential” used is not proper as the pCLB2-RFA1 *spp1*Δ cells did not die upon CPT. Instead, the word “critical” should be used.

We have replaced essential by critical.

- Page 10 Line 288: the same problem as Figure 6b—the author missed out showing the increase of Rad52 foci in *spp1*Δ cells was indeed caused by replication stress.

See our previous comments and the new Fig. 8a.

- Page 11 Line 313: In Fig 8b, what was the rationale for the author to choose “CPT, 30 min” as a replicative stress condition. If it is a right condition, should one expect a similar histone H3 ChIP profile in this situation as that in the Figure 7b?

We chose CPT 30 minutes because at this concentration it causes reversed fork without activation of checkpoint.

As mentioned in the general comment, in the revised version of the MS we now investigate the nascent chromatin organization (protection against MNase) upstream of the Tus/Ter barrier (and not at the vicinity of ARS in the presence of CPT). Recovered nascent DNA was analysed by qPCR using the same pair of primers used in the previous experiments (see next point).

- Page 11 Figure 8: The whole experiment setup should be in a genome-wide scale if the author expected to draw the final conclusion—Spp1 binding to chromatin is important for nascent DNA protection during RF stalling. The MNase-seq experiment should be pursued.

For the reasons mentioned in the general comment, we decided to perform the MNase experiment specifically at the Tus/Ter barrier. This experiment is now shown in the new Figure 7 (a, b, c). As described before, cells were arrested in G1 in a raffinose-galactose medium and released from the G1 arrest in the presence of EdU to allow its incorporation into nascent DNA. Cells were then collected at 20, 30 and 40 min and split into three parts for histone H3 ChIP, undigested chromatin and MNase-digested chromatin. Click reaction was used to conjugate biotin with EdU-labeled DNA enabling nascent DNA recovery by streptavidin pulldown. We performed streptavidin pulldown on both MNase-digested and sonicated (undigested) DNA to allow quantification of the protected nascent chromatin. Recovered nascent DNA was then analyzed by qPCR using the same pair of primers used in the previous experiments (see Figure 7a and text). In WT cells the chromatin protection decreased at regions upstream the Ter site at 30 and 40 min reflecting chromatin remodelling during replication fork stalling at the Tus/Ter barrier. Consistent with the results obtained in the *spp1*Δ mutant in the previous figures, we observe a strong decrease of chromatin protection in the absence of Spp1 (Fig. 7c). These results emphasize the role of Spp1 in protecting nascent chromatin during fork stalling at the barrier.

Minor points: • Some figure labels should be checked.

Thank you! We have checked accordingly.

- Page 6 Line 142 Checked Fig1 has been replaced by Fig. 2
- Page 6 Line 144 Checked
- Page 6 Line 156 Checked
- Page 7 Line 170: Checked (Fig. 3a has been replaced by Fig. 3a, b)

- Page 10 Line 288 Checked (This figure is now Fig. 8)

- Page 5 Line 107: RFB: replication fork barrier Corrected
- Page 5 Line 128: RF: replication forks Corrected
- Page 10 Line 280: RS: replication stress Corrected
- Page 7 Line 186: The sentence should be checked Corrected

- Page 8 Line 214: SPP1 mutants \diamond spp1 mutants Corrected

Reviewer #2 (Remarks to the Author):

This manuscript from the Geli laboratory reports that the Spp1 protein of *Saccharomyces* is involved in preventing DNA at stalled DNA replication forks from being processed by Exonuclease 1. Further, they show that Spp1 is recruited to chromatin through an interaction with histones carrying the H3K4me3 mark. There is considerable interest in understanding the steps that occur downstream of replication fork stalling and this study utilizes a neat system that creates a locus-specific block in the yeast genome. The results would be of interest to labs working on DNA replication, repair, and chromatin structure, but in its present form it has some weaknesses that I would recommend be addressed prior to publication.

Major Points

1. The standard of the English would benefit considerably from careful editing and proofreading.

We have proofread the manuscript and hopefully corrected the mistakes.

2. There are two issues with the 2D gels presented in Figure 1. First, the modest quality of the image makes it impossible to discern where the Y-arc and other characteristic structures lie. The X-shaped molecule is also poorly visible.

We have integrated a new figure with better quality (new Fig. 1b)

To make the data convincing, the authors should run gels with samples taken at different time points after release from alpha-factor, including much earlier time points than are shown, and couple these with flow cytometry traces at each time.

The first 2D gel experiment was performed under these conditions because we wanted to make sure that most of cells were in S phase. At earlier time points, a population of cells was still in G1 phase as revealed in multiple FACS profiles. We thus did the experiment at 25°C, because we observed a better synchrony at this temperature. However, decreasing the temperature and working in S-Raffinose dramatically slowed the RF progression.

We have performed new kinetics and collect samples at several time points, but the signals in the 2D-gels were very faint. In summary, the synchrony required to obtain a high number of cells in S phase to perform the 2D-gel is particularly laborious and delicate. Nevertheless, in the revised version of the MS, we have performed a new 2D gel analysis of cells grown in S-Raffinose at 30°C and collected WT and *spp1Δ* cells at 30 and 40 minutes. The data are shown in figure 4b. Unfortunately, as shown in the same Fig. 4b, the corresponding FACS profile shows that at 30 min, an important fraction of cells is still in G1. Nevertheless, we were able to show that the RF arrest induced by the barrier occurs earlier and is exacerbated in *spp1Δ* cells.

Second, it is claimed that the X-shaped molecules are recombination intermediates. This cannot be claimed without analysis of a *rad51* or *rad52* mutant.

We report the accumulation of X-shaped intermediates in 2D-gels, which could be attributed to replication fork reversal, recombination intermediates and, in this specific case, also to converging forks from the ARS306 origin. It was not our intention to formally claim that the X-spike is due only to *RAD51/52*-dependent recombination. In the text we replace the sentence (line 111):

" Importantly, we also detected a visible accumulation of X-shaped DNA intermediates that reflect DNA structures such as fork reversal and recombination intermediates."

with this sentence:

"We also detected a visible accumulation of X-shaped DNA intermediates, which could be attributed to replication fork reversal, recombination intermediates or converging forks arriving from the ARS306 origin"

3. The timing of appearance of the replication fork at the barrier seems unusually late. As the barrier is adjacent to ARS305, which fires immediately after S-phase entry, it is hard to

imagine that it takes 40 mins after alpha-factor release for replisome components to be detected at the barrier. What is the possible explanation for this?

Previous work on the Tus-Ter barrier has shown that the replisome is maximally arrested by about 30 mins, after which the transient arrest starts to be overcome. Is it possible that the authors have missed the peak of binding because of using time points that are too late? Please also see a related comment in point 8 below.

In our study, we used SD-Raffinose. Based on several FACS profiles, we found that cells do not enter S-phase until 20 min in WT when cultured in S-Raffinose, but instead enter S-phase with some asynchrony. This explains why the blocked fork is detected at 40 minutes in WT. The fact that the cells in G1 are not completely synchronous leads to a more diffuse Cdc45 signal between times 30, 40 and 50 minutes. Indeed, in 2D gels (new data shown in Figure 4b) we detect a diffuse signal from the blocked fork at 30 and 40 minutes in WT.

4. To make claims about the speed of the fork movement being faster in *spp1* mutants, it is necessary to do the analysis more rigorously by doing 2d gels probed with different probes from along the genomic region in a time course experiment. One possible explanation for the effects seen is not that the replisome speeds up, but instead that the *spp1* mutant cells enter S-phase more rapidly due their size. Any mutant cell that is unusually large tends to fulfill the G1 size checkpoint very quickly and race into S-phase after release from alpha-factor.

As mentioned above, in S-Raff (+GAL) WT cells start to enter S phase after 20-25 min and progress slowly. In the same medium, we observed that *spp1*Δ mutant cells grow better (still slower than in SD) and enter S phase more efficiently. This could explain why we detect Cdc45 earlier. As mentioned in our response to referee 1, we now provide DNA combing results suggesting that the replisome moves faster in S-Raff (+GAL).

5. Previous studies indicated that recombination mechanisms generated many of the Tus associated mutations. This dependence should be tested here too on the mutations in *spp1* cells.

Indeed, Larsen et al. (PNAS, 2017) previously reported that deleting *SGS1* in cells harbouring the TUS/*Ter* barrier induces X-DNA structures and increases the *URA3* mutation rate suggesting that Homologous Recombination Repair (HRR) contributes to the mutagenic process. Analysis of the *URA3* mutation rate in *rad51*Δ cells was complicated because deleting *RAD51* strongly increases the mutagenic rate even in the absence of TUS (Larsen et al., PNAS 2017). We have therefore analysed the *URA3* mutation rate in the absence of *RAD52* that eliminates all types of HRR in *S. cerevisiae*. To answer Referee's concern, we measured the *URA3* mutation rate in *rad52*Δ and *rad52*Δ *spp1*Δ cells. The results presented in the new Figure 6 show that in the absence of *RAD52*, the increase in mutation rate induced by TUS expression is promoted by a mechanism other than HRR. The fact that microdeletions (<3bp) are increased in *rad52*Δ cells suggests that NHEJ could operate in the absence of *RAD52* (as it might be also the case in the absence of *SPP1*). Interestingly, deleting both *SPP1* and *RAD52* have additive effects on the *URA3* mutation rate suggesting that two

mechanisms may operate in the double mutant. We discuss this possibility in the discussion of the revised manuscript.

6. There is a lack of consistency in how the cells are treated. In some experiments camptothecin is added, but then on page 12, HU is added. There was no explanation for the use of different compounds in these experiments.

We are more explicit in the revised manuscript. We choose to use CPT because it induces fork reversal which has been proposed to stabilize replication forks (Menin et al. EMBO Rep 2018). Our data indicate that RPA becomes critical to protect excessive ssDNA that accumulates at CPT lesions in *spp1Δ* cells. We further assessed *spp1Δ* sensitivity to different genotoxic stresses (CPT, HU, MMS) in combination with the *rfa1-D228Y* and *rfa1-t11* mutants. Our results (shown in Figure 8) indicate that a fully functional RPA is required in cells lacking Spp1 exposed to the three replication-stress-inducing agents.

7. Has it been tested whether induction of the barrier alone is sufficient to kill an *mph1 rad5 spp1* triple mutant, rather than using HU?

This experiment has been done. Induction of the barrier has no effect on the viability of the *mph1Δ rad5Δ spp1Δ* triple mutant suggesting that one stalled fork at the TUS/TER barrier is not enough to kill the triple mutant (not shown in the manuscript).

8. I found the EdU data in Fig 4c confusing. In the WT cells, there was only minor amounts of EdU incorporated before the 45 min time-point and yet the flow cytometry traces in Fig 3 indicate that the WT cells are in mid S-phase and in many cases even in G2 by at an earlier time-point (40 mins) after release.

The reason for this is that the way of performing kinetics and processing the cells is not the same. In Figure 3, a total of 5ug/ml of alpha factor was added while in EdU FACS 10 μg/ml alpha factor was added. The cells therefore take longer to exit from the G1 phase. This will be specified in the legend of figure 4c.

Minor Points

1. The authors should standardize on calling the complex COMPASS or Set1C. Both names are used throughout.

In the revised manuscript we refer to the complex as Set1C, except in the title (for indexing purpose)

2. There are several overstatements and conclusions that, while perhaps correct, are only one possible explanation. It would be preferable to see these toned down a bit. For example, on page 9 it is stated that 'Spp1 restricts nucleolytic degradation of nascent DNA'. I don't think it was shown that it is nascent DNA that is being degraded. Also, on page 9 'The mutation rate increase is one of the hallmarks of excessive ssDNA formation this, these results confirms that Spp1 leads to excessive ssDNA formation as RF stalls'. This sentence is ungrammatical and it is also an overstatement.

We agree. The whole paragraph has been rewritten.

REVIEWER COMMENTS

Reviewer #1 (Remarks to the Author):

In the revised manuscript, the author tried to emphasize that the conclusions would only be applied to the single, site-specific replication barrier (the Tus/Ter complex). These could include the conclusions drawn from figure 1 to 3, saying that the Set1C subunit Spp1 is recruited to the Tus/Ter complex and that the recruitment is independent of Set1C but depends on its PHD finger domain, recognizing the parental chromatin marks, H3K4me3. These results could be the base for an interesting concept about a novel function of Spp1 in stalled replication forks in general, which is exactly the author's attempts to conclude in figure 4 (FACS profile of the cell progression; FACS profiles with bivariate EdU), figure S4 (DNA combing results), and the whole figure 8 (Rfa1-CFP and Rad52-YFP foci; genetic results). The results of new analyses in figure 8 enable them to conclude that "Deleting SPP1 sensitizes cells to RPA dysfunction" because RPA has a general function in stabilizing single-stranded DNA (ssDNA) intermediates that arise when the replication machinery encounters obstacles such as DNA lesions or tightly bound protein complexes. Therefore, the importance of their results really lies in the indication that "Spp1 preserves genomic stability in the face of replication stress".

However, the revised manuscript still falls short of demonstrating this idea for two major reasons: 1) The implication of Spp1 generally being recruited to stalled replication forks was not be able to be demonstrated using genome-wide approaches, with the explanation that the genome-wide specific peaks of Spp1 in the experiments were very difficult to detect as the tight association of Spp1 to Set1C. However, the author had claimed that Spp1 is recruited to the stalled forks independently of set1C. This contradictory is necessary to be clarified to verify the main claims of these studies. In addition, the manuscript is missing important controls such as the SWD3 deletion strain (reflecting the functional roles of Set1C) in these experiments mentioned above, which are necessary to validate the conclusions of the author--- "Spp1's protection function, exerted independently of its interaction with Set1C, at the stalled folks."

2) The current work lacks mechanistic insight as to Spp1's role at the stalled forks. An attempt is made looking at H3 ChIP profiles and nascent DNA protection at forks stalled at the Tus/Ter barrier but it is far too preliminary and more data are required to support the proposed model (figure 9).

Reviewer #2 (Remarks to the Author):

I think that the authors have made a good effort to revise the manuscript in response to the referees' comments. I would recommend that the manuscript be accepted for publication now.

Point by point response to Reviewers 'concerns.

Reviewer #1 (Remarks to the Author):

In the revised manuscript, the author tried to emphasize that the conclusions would only be applied to the single, site-specific replication barrier (the Tus/Ter complex). These could include the conclusions drawn from figure 1 to 3, saying that the Set1C subunit Spp1 is recruited to the Tus/Ter complex and that the recruitment is independent of Set1C but depends on its PHD finger domain, recognizing the parental chromatin marks, H3K4me3. These results could be the base for an interesting concept about a novel function of Spp1 in stalled replication forks in general, which is exactly the author's attempts to conclude in figure 4 (FACS profile of the cell progression; FACS profiles with bivariate EdU), figure S4 (DNA combing results), and the whole figure 8 (Rfa1-CFP and Rad52-YFP foci; genetic results). The results of new analyses in figure 8 enable them to conclude that "Deleting SPP1 sensitizes cells to RPA dysfunction" because RPA has a general function in stabilizing single-stranded DNA (ssDNA) intermediates that arise when the replication machinery encounters obstacles such as DNA lesions or tightly bound protein complexes. Therefore, the importance of their results really lies in the indication that "Spp1 preserves genomic stability in the face of replication stress".

We thank Reviewer 1 for his/her positive comments.

However, the revised manuscript still falls short of demonstrating this idea for two major reasons:

1) The implication of Spp1 generally being recruited to stalled replication forks was not be able to be demonstrated using genome-wide approaches, with the explanation that the genome-wide specific peaks of Spp1 in the experiments were very difficult to detect as the tight association of Spp1 to Set1C. However, the author had claimed that Spp1 is recruited to the stalled forks independently of set1C. This contradictory is necessary to be clarified to verify the main claims of these studies. In addition, the manuscript is missing important controls such as the SWD3 deletion strain (reflecting the functional roles of Set1C) in these experiments mentioned above, which are necessary to validate the conclusions of the author--- "Spp1's protection function, exerted independently of its interaction with Set1C, at the stalled folks."

In this work, we show that Spp1 is recruited to the Tus/Ter barrier, while Swd3 is not. We also show that RNA-PolIII, which is involved in the recruitment of the Set1C complex, is not detected at the barrier either. We also show that Spp1 recruitment to the barrier depends on its PHD domain, which is able to recognize H3K4me2 and H3K4me3. This is somewhat analogous to the situation observed during the first meiotic division, in which Spp1 is recruited to meiotic break sites through the interaction of its PHD domain with H3K4me3, independently of RNA-PolIII and Set1C. We see no contradiction in asserting that Spp1 is recruited to blocked forks independently of Set1C, whereas it is recruited genome-wide by the Set1C complex to the very many binding sites of the Set1C complex.

We don't think that analysis of the *swd3Δ* mutant is the right control to demonstrate that Spp1 is recruited to the barrier independently of Set1C, since in the *swd3Δ* mutant H3K4 methylation (mono, di, tri) is abolished. This will affect per se the interaction between the Spp1 PHD domain and the nucleosomes that will be unmethylated at H3K4. This will not mean that the interaction of Spp1 at the barrier depends on Swd3. This point is now underlined in the discussion and we have mitigated the conclusion by saying that we cannot be 100% sure that

Spp1 recruitment to the barrier is independent of the Set1 complex (see in the discussion lines 392-401).

2) The current work lacks mechanistic insight as to Spp1's role at the stalled forks. An attempt is made looking at H3 ChIP profiles and nascent DNA protection at forks stalled at the Tus/Ter barrier but it is far too preliminary and more data are required to support the proposed model (figure 9).

We believe that the new experiments presented in the new Figure 7 of the revised version, in which we performed streptavidin pulldown on both MNase-digested DNA and sonicated (undigested) DNA to enable quantification of protected nascent chromatin, are the best that can be done to provide mechanistic information on the role of Spp1 at the fork stalled at the Tus/Ter barrier.

Reviewer #2 (Remarks to the Author):

I think that the authors have made a good effort to revise the manuscript in response to the referees' comments. I would recommend that the manuscript be accepted for publication now.

We thank Reviewer 2 for his/her recommendation.